# Alternative splicing regulation in plants by SP7-like effectors from symbiotic arbuscular mycorrhizal fungi

Ruben Betz [1,3], Sven Heidt[1,3], David Figueira-Galán [1], Meike Hartmann[1], Thorsten Langner [2] & Natalia Requena [1] ✉

Most plants in natural ecosystems associate with arbuscular mycorrhizal (AM) fungi to survive soil nutrient limitations. To engage in symbiosis, AM fungi secrete effector molecules that, similar to pathogenic effectors, reprogram plant cells. Here we show that the Glomeromycotina-specific SP7 effector family impacts on the alternative splicing program of their hosts. SP7-like effectors localize at nuclear condensates and interact with the plant mRNA processing machinery, most prominently with the splicing factor SR45 and the core splicing proteins U1-70K and U2AF35. Ectopic expression of these effectors in the crop plant potato and in *Arabidopsis* induced developmental changes that paralleled to the alternative splicing modulation of a specific subset of genes. We propose that SP7-like proteins act as negative regulators of SR45 to modulate the fate of specific mRNAs in arbuscule-containing cells. Unraveling the communication mechanisms between symbiotic fungi and their host plants will help to identify targets to improve plant nutrition.

In nature, plants engage in multiple microbial interactions that ultimately determine plant health and productivity. During these associations, both plants and microbes exert all their capabilities to take control of the interaction, including the delivery of peptides, proteins and small RNAs to the other partner in order to manipulate their cellular program[1–4]. These effector molecules are key components of the interaction between microbes and their host plants. Effectors allow microbes to remain invisible to the surveillance system of plant cells, to subvert defense reactions and/or to manipulate plant metabolism. Although most effectors characterized belong to pathogenic microbes, increasing evidence shows that beneficial associations also rely on the delivery of effector molecules to shape the mutualism[5,6].

AM symbiosis, involving fungi from the Glomeromycotina subphylum, is the most widespread plant-fungal association[7]. These obligate symbiotic and biotrophic fungi provide plants with essential mineral nutrients, primarily phosphate, in exchange for fixed carbon that is exchanged at tree-like structures located in cortical cells called arbuscules[8]. This complex symbiosis requires an extensive signal exchange between the plant and the fungus and we predicted that effectors could play a central role in this communication[9]. The increasing availability of sequenced genomes from different AM fungal species and prediction tools has revealed the existence of many putative effector proteins[10–14]. However, only a handful of them have been functionally characterized and shown to play a role in the symbiosis[9,15–18]. Microbial effectors have a wide range of possible final locations *in planta*, ranging from the apoplastic space to different intracellular compartments including the nucleus. This allows effectors to find their specific targets and control multiple cellular processes[19–21]. Nuclear effectors have been shown to modulate hormonal pathways, host metabolism or immune responses by different mechanisms that include interactions with nuclear proteins, DNA or RNA[19,21]. Interestingly from the five effectors characterized in the fungus *Rhizophagus irregularis*, three are located in the nucleus[9,16,18]. This might reflect the large number of predicted nuclear effectors in this fungus, as revealed in an analysis of *R. irregularis* secretome during symbiosis, in which from ca. 300 putative effectors, almost one third

[1]Joseph Kölreuter Institute for Plant Sciences. Molecular Phytopathology Department, Karlsruhe Institute of Technology (KIT) - South Campus, Fritz-Haber-Weg 4, Karlsruhe, Germany. [2]Max Planck Institute for Biology Tübingen - Max-Planck-Ring 5, Tübingen, Germany. [3]These authors contributed equally: Ruben Betz, Sven Heidt. ✉e-mail: natalia.requena@kit.edu

had a predicted nuclear localization signal (NLS)[17]. This points the nucleus as a key target for arbuscular mycorrhizal symbiotic effectors. This is not unusual, thus for instance, two of the few characterized effectors from rhizobia, ErnA and Bel2-5 are also targeted to the plant nucleus and shown to promote nodulation[5]. But also, numerous effectors of plant pathogens have the nucleus as their playground[19,20].

From all nuclear-targeted effectors characterized so far, only a few have been shown to impact pre-mRNA splicing. This is remarkable because alternative splicing (AS) is a key regulatory mechanism of gene expression in plants[22–25], where it is estimated that between 60–70% of intron containing genes undergo AS[26]. In the crop plant potato from *ca*.33.000 genes more than 44.000 transcripts are predicted, and 7000 genes are known to have more than one transcript (Phytozome v.13). Alternative splicing in plants contributes to the ability of plants to respond to environmental changes including abiotic and biotic stresses, but also to control developmental processes[22,24,27]. Furthermore, the role of AS in plant immunity is increasingly recognized with defense responses altered in mutants of splicing regulators[28–30], altered splicing landscape during infection by pathogens[31–33], or the discovery of effectors targeting AS[33–36].

Pre-mRNA splicing occurs at the spliceosome, where besides the large ribonucleoprotein complex (U1, U2, U4, U5 and U6) many other accessory proteins participate. Among those, the serine-arginine (SR) rich proteins are a family of RNA-binding proteins with a prominent role in constitutive and alternative splicing. They were first identified as chromatin interacting proteins, and have been shown to participate not only in pre-mRNA splicing but also in mRNA nuclear export and translation control[37,38]. SR45 and its human counterpart RNPS1 are members of the conserved Transformer-2 subfamily within the SR superfamily. They are unusual splicing factors containing two arginine-serine (RS) rich domains flanking an RNA recognition motif (RRM)[39]. While the RRM motif determines RNA-binding specificity, the RS domains participate in protein-protein interactions[40]. RNPS1 is a peripheral protein of the exon-junction-complex (EJC) that serves as platform for splicing regulatory complexes to maintain transcriptome surveillance by regulating alternative splicing and nonsense mediated decay (NMD)[41,42]. In addition, RNPS1 and more recently SR45 as components of the apoptosis and splicing-associated protein (ASAP) complex have been shown to modulate co-transcriptional repression[43–46]. In *Arabidopsis*, SR45 was also postulated as part of the EJC[47] and shown to colocalize with U2AF[35]b, Cactin and other SR proteins in nuclear condensates[40,48]. *Arabidopsis* SR45 is even able to complement the splicing of some pre-mRNAs in splicing-deficient HeLa cell S100 extract[49]. A loss of function of the SR protein SR45 led to the surprising finding that this protein, besides its already known roles in development and abiotic stress[39,49,50], was a suppressor of innate immunity in *Arabidopsis*[28].

Here we demonstrate that beneficial symbiotic AM fungi employ an effector protein family, widely conserved across the Glomeromycotina, that interacts with components of the plant alternative splicing machinery, most significantly with SR45, and modify the alternative splicing of a subset of genes when expressed *in planta*.

## Results
### The SP7-like effector family
SP7 belongs to a small family of proteins in the symbiotic arbuscular mycorrhizal fungus *R. irregularis* that shares a conserved modular structure, comprising a signal peptide, a series of hydrophilic imperfect repeats and, in several cases, a nuclear localization domain (Fig. 1a). Based on the previously identified sequences[9–11], we searched on the new annotation of the *R. irregularis* genome[12,13] and identified five *core* RiSP7-like sequences, RiSP7, RiSP2, RiSP5, RiSP31 and RiSP1 (Fig. 1a). Surprisingly, for four of them there are several highly conserved paralogues scattered through the genome in different chromosomes, with the exception of *RiSP2*. Only in two cases, more than

one *SP7-like* gene is found in the same chromosome (Fig. 1b). Paralogues encode proteins that share between 85% and 96% amino acid identity (Supplementary Fig. 1), while the identity among the five core RiSP7-like proteins is no more than 49% (Supplementary Fig. 1). To find out if SP7-like effectors and their genomic location are conserved between different strains of *R. irregularis*, we carried out BLASTN-based sequence homology searches in available assemblies of both, homo- and heterokaryotic strains[12,13,51]. SP7-like effectors are highly conserved in the investigated homokaryotic strains. We only found presence/absence variation for *RiSP1* in strain C2, which lost two copies (Source data). Conversely, we noted extensive presence/absence variation mainly in the form of gene losses in heterokaryons (Source data), with the exception of strain A5 which only lost one copy of *RiSP5* in both nuclei (Supplementary Fig. 2). In heterokaryotic strains, each genome can differ in gene content and expression[12]. For SP7-like genes, however, we observed frequent gene loss that often affected both nuclei (Source data). Two genome compartments have been described for *R. irregularis*[12], with an A compartment more gene dense and higher expressed, and a B compartment with higher repeat density and enriched in candidate effectors[51]. The *RiSP7* gene was localized in the B compartment[12]. We checked the localization of the whole family of SP7-like effectors in the heterokaryon A4 and could show that with one exception all of them are located in compartment B (Supplementary Fig. 3).

Because the AM symbiosis is formed by many different fungal species, we investigated whether similar effectors would exist in other AM fungi (Supplementary Data 1 and 2). Amplification of two putative homologs from *Gigaspora margarita* and *Gigaspora rosea*[52,53] and alignment of the proteins with the SP7-like sequences from *R. irregularis* revealed a similar structure and the presence of a characteristic signature that defines the family (Fig. 1a, Supplementary Figs. 1 and 4). Furthermore, a conserved intron preceding the first repeat was found in all *SP7-like* genes (Supplementary Fig. 4) including also other species such as *R. diaphanus* and *R. clarus*, suggesting a common gene ancestor for this effector family despite their sequence divergence (Fig. 1c). The exon-intron structure has been used to identify a common origin for other effectors, including the Y/F/WxC effector family from the pathogenic fungus *Blumeria graminis* f.sp. *hordei*[54] or the Bicycle effectors from aphids[55]. Using the criteria of this conserved signature and the conserved intron, we found that SP7-like proteins are present in the Glomerales and Diversisporales, but there are so far no hints of them in more ancient orders such as Paraglomerales or Archaeosporales nor in other subphyla from the Mucoromycota such as Mortierellomycotina or Mucoromycotina (Fig. 1d and Supplementary Data 1 and 2). Taken together, the conservation of the SP7-like family within the Glomeromycotina and its expansion in *R. irregularis* suggests that they could play an important role in the AM symbiosis.

### SP7-like proteins localize to the plant nucleus in nuclear condensates
We had shown previously that RiSP7 localizes in the nucleus of the plant when expressed *in planta* as predicted by the presence of an NLS signal[9]. However, several of the SP7-like members of the gene family, including those from *Gigaspora* species, apparently lack a clear NLS. Therefore, in order to investigate their localization, we expressed several of them ectopically in *Nicotiana benthamiana* epidermal cells, fused to GFP either N-terminally or C-terminally. Surprisingly, all of them showed predominantly nuclear localization despite lacking a predicted NLS (Fig. 2a). An exception was RiSP31 that only localized to the nucleus when carboxy-terminally labeled, albeit in a very low amount, despite possessing a distinct NLS. All analyzed SP7-like effectors were also present in the cytoplasm to a certain extent, with the exception of RiSP2 that was solely located in the nucleus (Fig. 2b). Furthermore, it could be shown that in some cases RiSP7 and RiSP5 localized to P (processing) bodies in the cytoplasm, as confirmed by

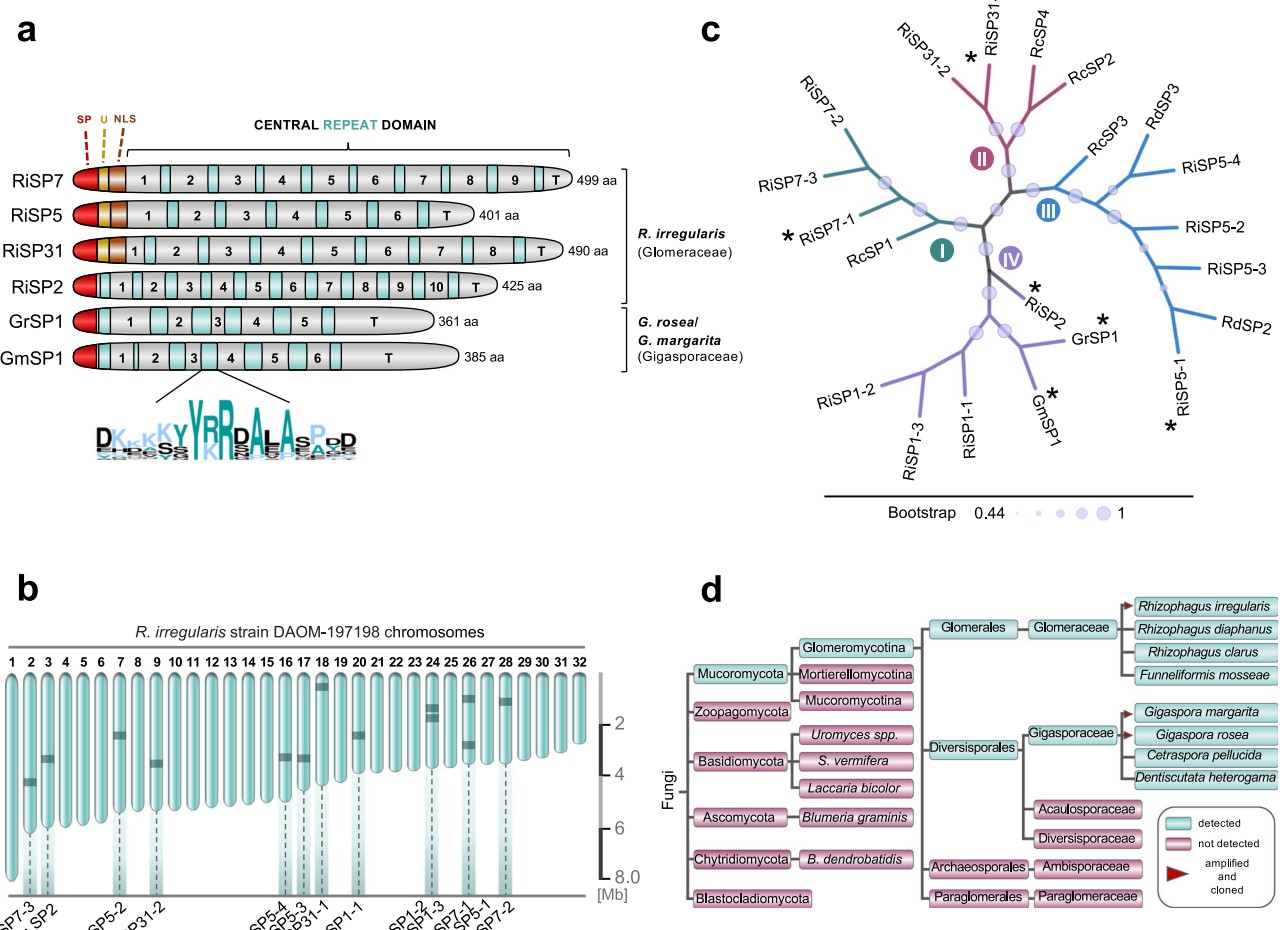

**Fig. 1 | The multi-membered SP7 effector family is conserved and specific to AMF. a** Protein features of cloned SP effectors. All effectors possess a signal peptide (SP, red, SignalP-5.0). For some members this is followed by a conserved 10–12 aa stretch of unkown function (U, yellow) and a consecutive bipartite NLS (brown, PSORTII). The repeat domain contains 6–11 imperfect hydrophilic repeat units. Variable aa (insertions/deletions/substitutions) define each repeat-subunits. A characteristic highly conserved island marks the end of each repeat unit (green colored boxes, consensus logo). Terminal repeats (T) lack these islands.
**b** Distribution of SP7 family effectors in the genome of *R. irregularis*. Chromosomes 1–32 of *R. irregularis* strain DAOM-197198 are depicted according to Manley et al. Relative positions of SP7 family members are indicated by dark bars. *R. irregularis* contains four core members (*SP7-1, SP1-1, SP2, SP5-1* and *SP31-1*, asterisks) that exist as paralogs in varying numbers. **c** Phylogenetic relationship of full length SP7 family

members identified for different AMF species. Shown is an unrooted Maximum Likelihood tree (MEGA11, JTT modeling) of SP7 family members after ClustalW alignment. Bootstrap values (1000 replicates) are shown by branch circles. SP7-like effectors cluster into four distinct clades (I-IV). Tree was visualized using iTOL. Asterisks mark cloned effector members. Ri (*Rhizophagus irregularis*), Rc (*Rhizophagus clarus*), Rd (*Rhizophagus diaphanus*), Gm (*Gigaspora margarita*), Gr (*Gigaspora rosea*). **d** Analyzes of genome and RNAseq data (NCBI Sequence Read Archive (SRA)) using SP7 family effector sequences as query revealed the ubiquitous distribution of SP effector genes or transcripts in the subphylum Glomeromycotina, while they were not found in any other of the analyzed fungal taxa (see Supplementary Data 1 and 2 for details). Red triangles mark species that were used to clone full length genomic or cDNA effector sequences.

the co-localization with the P body marker AtDCP2 (Supplementary Fig. 5). A higher magnification of the nucleus showed that often SP7-like proteins accumulated in nuclear bodies in inhomogeneous numbers and sizes, but never in the nucleolus (Fig. 2a and Supplementary Fig. 6). Stable transformed plants of *Arabidopsis thaliana* expressing *RiSP2* showed localization in nuclear condensates of root and root hair cells, further supporting that this is the most frequent localization of these proteins as observed in *N. benthamiana* (Supplementary Fig. 7). Different types of nuclear bodies have been identified forming as a result of phase separation processes, often involving the recruitment of intrinsically disordered region-containing proteins[20]. Phase separation is mediated by molecules with the ability to form many interactions allowing the formation of subcellular or intra-organelle membrane-less compartments termed biomolecular condensates[56,57]. From those, nuclear condensates have a recognized role as mRNA processing organelles. All SP7-like predicted proteins are almost entirely intrinsically disordered (Supplementary Fig. 8), suggesting the

possibility that they might have multiple interaction partners and be components of biomolecular condensates.

## SP7-like effectors highjack the mRNA processing pathway
In order to identify interaction partners of SP7-like effectors, we carried out yeast two hybrid (Y2H) assays using them as baits against libraries from several plants. Surprisingly, given the low overall sequence similarity among the fungal effectors, their Y2H interactome revealed a striking convergence on proteins related to mRNA processing, RNA binding and nuclear speckle categories, as revealed by gene ontology enrichment (Supplementary Fig. 9 and Supplementary Data 3). These results are in line with the localization pattern observed in nuclear condensates and suggests that SP7-like effectors could be impacting on the fate of specific mRNAs. To obtain *in planta* evidence for this hypothesis, co-immunoprecipitation assays in *Nicotiana benthamiana* expressing GFP-tagged RiSP7 were carried out. The results not only validated several of the interaction proteins identified in the

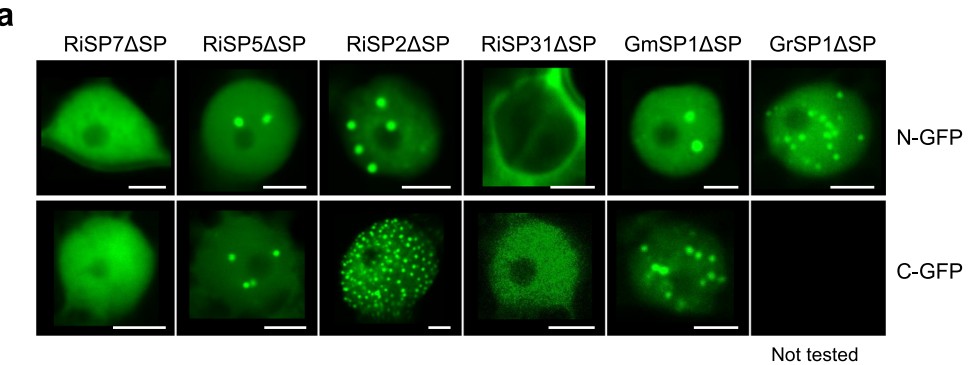

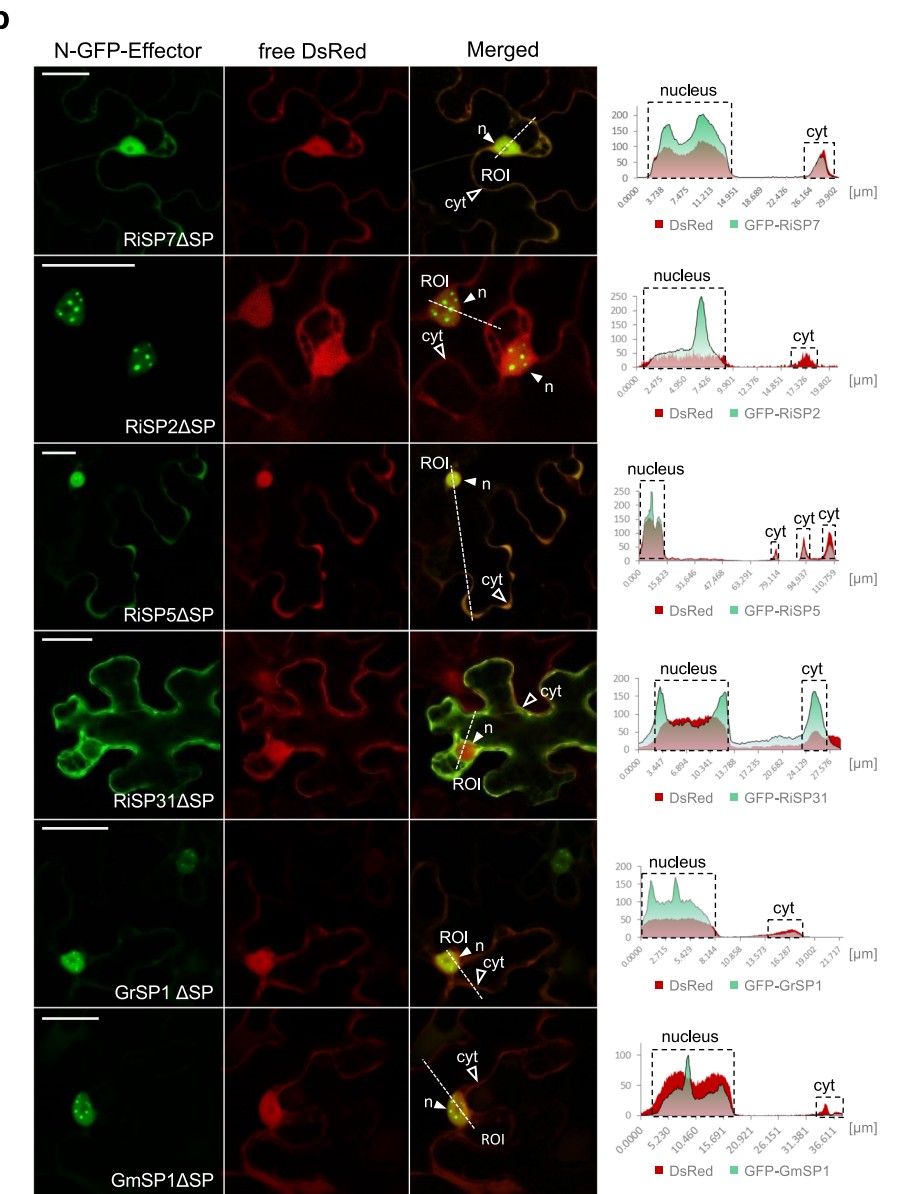

Y2H analysis, but also revealed further components of the mRNA processing machinery from splicing to translation control (Supplementary Fig. 9, Supplementary Data 3).

Interestingly, among SP7-like interactors, several SR proteins and their regulators were identified (Supplementary Data 3 and Supplementary Figs. 9, 10 and 11). Among those, the serine-arginine rich protein SR45 was consistently retrieved as interaction partner of all

SP7-like proteins. SR proteins are often in multiprotein complexes regulating the different steps of mRNA life. For example, SC35, a central component that links SR proteins to co-transcriptional splicing[58] was also identified as an interactor of RiSP2. This result suggests that SP7-like effectors could assemble with the spliceosome already during transcription. In support of this hypothesis, the largest subunit of the RNA polymerase II was immunoprecipitated with RiSP7

**Fig. 2 | The SP7-like effector family localizes to the plant nucleus and accumulates in nuclear condensates. a, b** Subcelluar localization of SP7-like effectors lacking their SP (ΔSP) fused with either their N-terminus (N-GFP) or C-terminus (C-GFP) to eGFP under control of the P35S promoter after transient expression in *N. benthamiana* epidermal leaf cells. **a** Shown are localization patterns in single plant nuclei. All but one effector target the plant nucleus. While RiSP7ΔSP exhibited a homogeneous nuclear distribution, RiSP2ΔSP, RiSP5ΔSP and the *Gigaspora* orthologs additionally localized at nuclear condensates. As exception, RiSP31ΔSP was mainly present in the surroundings of the nucleus (N-GFP) or detected only weakly in the nucleus (C-GFP). Scale bars represent 5 μm. **b** Representative overview pictures of SP7-like effector localizations (N-GFP) within the cellular context. As control, DsRed was co-expressed visualizing the plant nucleus and cytoplasm. All SP7 family effectors, with the exception of RiSP31ΔSP, are visible in plant nuclei (*n*, filled white arrowhead). In addition, RiSP7ΔSP, RiSP5ΔSP, RiSP31ΔSP and the *Gigaspora* proteins are present in the cytoplasm with varying intensities (cyt, blank arrowhead indicates exemplary region). No cytoplasmic signals can be detected for RiSP2ΔSP. ROI (Region of Interest, merged pictures) indicates the starting points of transection lines used for fluorescence intensity measurements. Right: Fluorescence intensity blots along transection lines individually generated for GFP and DsRed channels. Nuclear areas and cytoplasmic signals at the cell periphery are boxed. Maximum fluorescence intensity = 255. **Note:** Only presence and changes in intensity at specific positions can be compared between two different fluorophores. Overall intensity levels can differ. Scale bar = 50 μm.

(Supplementary Data 3). Interestingly, the RNA binding protein HRLP, that together with SR45 inhibits the co-transcriptional splicing of the flowering repressor FLC, was also identified as interaction partner of RiSP7. HRLP leads to R-loop formation and transcription inhibition, thereby promoting flowering[45].

Further support of the role of SP7-like effectors in mRNA splicing was obtained by the identification of two elements of the NineTeen Complex (NTC), MAC3a/Prp19 and MAC5a (Supplementary Data 3 and Supplementary Fig. 9). The NTC is a conserved eukaryotic protein complex that associates with the spliceosome and participates in splicing. In plants, it was shown to play a role in immunity as well as in developmental processes such as flowering[30,45]. Most interestingly, an interplay between RNA splicing and polyadenylation was also shown to play a role in immunity[30]. Here, we show that core components of the alternative polyadenylation pathway, such as CPSF30 or the cleavage stimulation factor CstF77 were also identified in the interactome of SP7-like effectors (Supplementary Data 3 and Supplementary Fig. 9). Altogether, these results suggest a picture where SP7-like effectors are recruited to nuclear condensates where they interact with components of the mRNA splicing machinery during transcription.

### SR45 is a core interactor of SP7-like effectors

Among all proteins identified in the interactome, the SR protein SR45 was retrieved by all SP7-like effectors and was also immunoprecipitated by RiSP7 (Supplementary Data 3). In contrast, validation experiments using Y2H and BiFCs assays showed that most of the identified SR and mRNA processing proteins are specifically interacting with some of the effectors but not with all of them (Supplementary Fig. 12). Using targeted Y2H we could show that all SP7-like proteins interacted with the SR45 orthologues from three different plants (*Medicago*, *Arabidopsis* and *Nicotiana*), and even with the two splice variants existing in *A. thaliana* (Fig. 3a). This is in agreement with the high degree of conservation of the plant SR45 proteins (Fig. 3b). MtSR45 from *M. truncatula* localizes in nuclear condensates (Fig. 3c) as it was previously shown for AtSR45[59]. Furthermore, both orthologues localize to the same condensates when co-expressed, while free eGFP does not (Fig. 3d). Bimolecular fluorescence complementation (BiFC) demonstrated that interactions between SP7-like effectors and MtSR45 also occurred in nuclear condensates (Fig. 3e). This was somewhat surprising, because while all SP7-like proteins showed a distinct nuclear localization when expressed alone, ranging from diffuse localization to accumulation at discrete nuclear bodies (Fig. 2a and Supplementary Fig. 6), co-expression with MtSR45, relocalized all SP7-like proteins to the same nuclear condensates occupied by MtSR45 or NbSR45 (Fig. 3f and Supplementary Fig. 13). This was not the case for the control protein GFP nor it is the case for all effectors that localize to nuclear condensates. We had previously shown that the *R. irregularis* effector CRN1 localizes to nuclear condensates different from those containing SR45[16] and Supplementary Fig. 13. SR45 as well as RNPS1 contain two intrinsically disordered domains (Supplementary Fig. 8) which are thought to enable flexible interactions with different protein and RNA partners in a concentration-dependent manner[60,61]. Hence, effector re-localization might be triggered by an increased nucleating activity of MtSR45 when present at higher concentrations due to overexpression. However, it has been shown that expression of *AtSR45* under its natural promoter also drives its localization to nuclear condensates thus suggesting that nuclear condensate localization per se might not be due to an overexpression artifact[62]. Furthermore, not all SR proteins when overexpressed form nuclear condensates nor recruit the *R. irregularis* effector to their localization if co-expressed, as it is the case of the SR protein MtSR30 (Supplementary Fig. 13). But because condensates formed by intrinsically disordered proteins are concentration dependent[56], we cannot exclude that the size of these condensates in cortical cells during the symbiosis might be different.

To map the SR45 domain required for the interaction with SP7-like effectors, we next analyzed the ability of truncated versions of MtSR45 to interact with RiSP7 in a Y2H screen. The results showed that the carboxy terminal RS2 domain of MtSR45 is necessary and sufficient to mediate the interaction with RiSP7, while the RNA binding and the RS1 domain are dispensable (Fig. 4a). This result was confirmed for all SP7-like proteins and corroborated by BiFC assays (Figs. 4b, c and Supplementary Fig. 14). This finding is interesting, because the RS2 domain had been shown to be necessary and sufficient to drive localization of AtSR45 to nuclear condensates[63]. We showed that this is also the case for the *M. truncatula* orthologue (Supplementary Fig. 14). Furthermore, SP7-like effectors only fully co-localize with MtSR45 if the RS2 domain is present and the protein forms nuclear condensates (Fig. 4d and Supplementary Fig. 14).

As shown above, SP7-like protein effectors share only a low similarity regarding their overall sequence identity, but all follow the same modular protein structure. All the repeat subunits, except the truncated last repeat, end in a small stretch of highly conserved amino acids (Fig. 1a, Supplementary Fig. 1a). Given that the number of repeats in each of the SP7-like proteins is variable, we next asked the question of how many protein repeats were required for the interaction with SR45. To answer that, truncated versions of the RiSP7 protein containing one, two or three repeats were tested for their ability to bind MtSR45. When the interaction was analyzed by Y2H, a minimum of three repeats were required, but this interaction was not as strong as with the full-length RiSP7 protein (Fig. 4e). Experiments in *Nicotiana* showed that full co-localization with MtSR45 in nuclear condensates was only observed with the full-length protein, while three repeats were sufficient to show a partial co-localization, supporting the Y2H results (Fig. 4f). This is reminiscent of the situation of the RNA binding protein HRLP, that requires all of its intrinsically disordered domains to localize in nuclear condensates and to act as a flowering regulator[45]. Surprisingly, a two repeats construct of RiSP7 was sufficient to allow BiFC (Fig. 4g and Supplementary Fig. 15), suggesting that the interaction between RiSP7 and MtSR45 could be mediated by small linear motifs between each RiSP7 repeat and the RS2 domain of MtSR45, and that the strength of the interaction increases with the number of repeats (Supplementary Fig. 15).

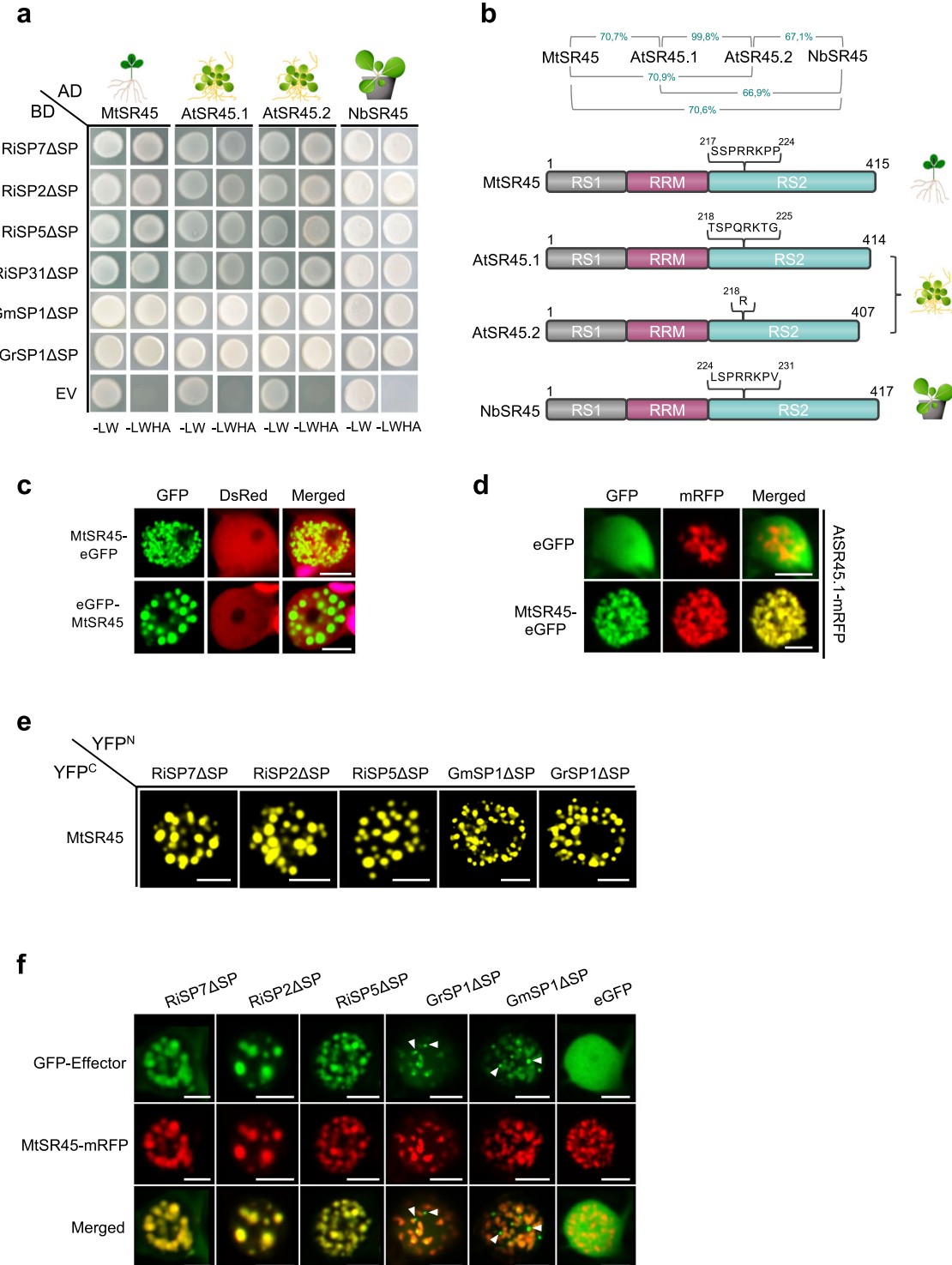

### The splicing proteins U2AF$^{35}$b and U1-70K are also core interactors of SP7-like effectors

*Arabidopsis* SR45 (Fig. 5a) was originally discovered as an interacting protein of the splicing factor U1-70K and later also shown to interact with the splicing factor U2AF$^{35}$b[40,64]. U2AF$^{35}$b was already retrieved as interacting partner of RiSP7 when the effector was first identified[9] and now in our interactome here. Both splicing factors were shown not only to interact with AtSR45 but also to co-localize in the nucleus[40,63]. MtSR45 did also interact with both splicing factors in yeast and in *Nicotiana* and localized to the same nuclear condensates

(Supplementary Fig. 16). We then tested whether SP7-like effectors were able to directly interact with both splicing factors. Indeed, this was the case, and all effectors interacted with MtU1-70K and MtU2AF$^{35}$b, both in Y2H as well as in *Nicotiana* assays (FigS. 5b, c). Interestingly, the reconstituted YFP signal in BiFC assays between RiSP7-MtU1-70K and the RiSP7-MtU2AF$^{35}$b interactions was observed as a nucleoplasmic signal, which was often accompanied by a low number (usually one to four) of nuclear foci (Fig. 5c). In contrast, the interactions between all other SP7-like effectors was observed mainly in nuclear condensates (Fig. 5c). This suggests, that in contrast to

**Fig. 3 | The SP7 effector family interacts with plant SR45 orthologs in nuclear condensates. a** Direct yeast two hybrid interaction assays showing that all SP7-like effectors (Baits, BD) interact with plant SR45 orthologues from *M. truncatula*, *A. thaliana* and *N. benthamiana* (Preys, AD). EV = Empty vector. Positive interactions are indicated by colony growth on media lacking leucine, tryptophan, histidine, adenine (-LWHA). **b** Percentage protein identity (top) and protein structures of SR45 plant orthologs. SR45 proteins consist of a RS1, RRM and a RS2 domain. *AtSR45* codes for the two isoforms AtSR45.1 and AtSR45.2 (Zhang and Mount, 2009). **c–f** Shown are localization patterns in single plant nuclei after transient expression of fluorescent tagged proteins under control of the P35S promoter in *N. benthamiana* epidermal cells. Scale bars represent 5 μm. **c** Subcellular localization of MtSR45 (N- or C-terminal eGFP fusion) in different shaped nuclear speckles. As control, free DsRed was co-expressed. Chloroplast autofluorescences are pseudo-colored in magenta (merged pictures). **d** Co-localization of MtSR45 C-terminally fused to eGFP and AtSR45.1 C-terminally fused to mRFP. MtSR45 and AtSR45.1 co-localize in nuclear speckles. As control, free expressed eGFP does not accumulate in speckles. **e** Bimolecular fluorescence complementation assays to confirm yeast two hybrid results from (**a**). Reconstituted YFP fluorescence was observed in nuclear condensates for plants co-expressing MtSR45 (fused to C-terminal half of YFP, YFP^C) with all tested SP7-like effectors (fused to N-terminal half of YFP, YFP^N). **f** Co-localization of SP7-like effectors N-terminally fused to eGFP and MtSR45 fused C-terminally to mRFP. In contrast to the eGFP control, all tested SP7-like effectors re-localized to nuclear condensates occupied by MtSR45. For *Gigaspora* effectors, exclusive nuclear bodies were observed in GFP channels (white arrowheads). **a-f** All experiments were carried out with SP7-like effectors lacking their signal peptides (ΔSP).

MtSR45, the splicing proteins MtU1-70K and MtU2AF[35]b have a weaker effect as nucleating factors.

## SP7-like effectors phenocopies the developmental defects of the AtSR45 mutant

*Arabidopsis* SR45 loss of function mutant *sr45-1* displays several developmental phenotypes, including a delay in flowering, smaller and narrow leaves, altered numbers of petals and sepals and reduced growth[49]. Given that all SP7-like effectors were able to interact with SR45 in all plants analyzed, we wonder whether their ectopic expression in planta could impact on these phenotypes. In order to investigate that, we expressed *RiSP7* and *RiSP2* constitutively in *A. thaliana* plants (Supplementary Fig. 18). Four homozygous lines of SP7 and one of SP2 were obtained. Despite the variability among the SP7 lines, we could clearly observe that they all, as well as the SP2 line, had a late flowering phenotype as compared to the wild type (Figs. 6a–d) that correlated with a higher expression of the flowering repressor *AtFLC* and a repression of its two downstream targets *AtFT* and *AtSOC1*[65] (Fig. 6d). In addition, *RiSP7* expressing *Arabidopsis* plants showed occasionally abnormal number of petals (Fig. 6c). This is also the case for the *sr45-1* mutant[49] and suggests that the flowering phenotype of SP7-like expressing plants could be regulated in an FLC-dependent manner. Recently, it has been shown that the RNA binding protein HRLP interacts with SR45 in nuclear condensates to regulate flowering time in *A. thaliana* by regulating splicing and transcript levels of *FLC*[45]. Consequently, the *hrlp* mutant has a reduced flowering[45]. *RiSP7 Arabidopsis* plants also showed higher splicing efficiency for some *AtFLC* introns but not for all, as shown for the *hrlp* mutant, what could result in a higher *AtFLC* expression level (Supplementary Fig. 18). Furthermore, SP7-like plants, as well as the *sr45-1* and the *hrlp* mutants showed stunted growth (Figs. 6a and e). These results led us to hypothesize that during symbiosis SP7-like proteins could act as repressors of SR45 activity in specific cells and on specific transcripts.

## SR45 is a negative regulator of the mycorrhiza symbiosis

*RiSP7* has been shown to be induced *in planta* during symbiosis[9,66], and more recently, using single-nucleus sequencing of mycorrhizal plants, to be enriched in the cluster of cells containing arbuscules[67]. Therefore, we hypothesized that its main function is exerted in those cells. To challenge the hypothesis that during mycorrhiza symbiosis RiSP7 could be impacting arbuscule development or function by repressing SR45, we ectopically expressed *MtSR45* in hairy roots of the symbiotic competent plant *M. truncatula* and analyzed the outcome of the symbiosis. To that end, we used a mycorrhiza-specific promoter which is only expressed in arbuscule-containing cells. As shown in Figs. 6f, g, plants overexpressing *MtSR45* in mycorrhizal plants had lower levels of fungal colonization with a reduction in the number of hyphae as well as of arbuscules per root. But more strikingly, almost all mycorrhization marker genes

analyzed, including functional genes for phosphate and carbon exchange, were downregulated when *MtSR45* was overexpressed in arbuscule-containing cells (Fig. 6h). Thus, it appears that SR45 is a negative regulator of mycorrhizal symbiosis that could be post-translationally repressed by SP7-like effectors in arbuscules during symbiosis.

## SP7 ectopic expression induces alternative splicing in potato

Given the central role of MtSR45 and MtU1-70K and MtU2AF[35]b as core interactors of SP7-like effectors, it seems plausible to think that this family of effectors might be able to alter the fate of specific target mRNAs by modulating their alternative splicing or of their upstream regulators. To challenge this hypothesis, we constitutively expressed *RiSP7* in potato plants and analyzed its effect on alternative splicing in roots. Potato plants expressing *RiSP7* showed a reduced growth phenotype, as also observed in *Arabidopsis* plants that expressed *RiSP7* or *RiSP2* (Fig. 6), with less shoot branches and larger leaves (Supplementary Fig. 17). Expression of *RiSP7* led to changes in gene expression of *circa* 4% of the potato genes in roots (DEGs = 1227) and to a much smaller number of genes that were subjected to alternative splicing events (DAS = 96) among the landscape of AS genes (7161) predicted in potato (Phytozome v13) (Supplementary Data 4). This is interesting because it reveals a specific selection of targets by the RiSP7 effector, in contrast to other effectors, such as the *Phythophthora* protein PSR1 that is able to cause global changes in AS affecting more than 5000 genes[68].

From the 96 DAS (differential alternative splicing) genes identified in response to the ectopic expression of *RiSP7*, only eight were also differentially expressed with seven of them repressed (Supplementary Data 4). This suggests that these AS forms could be subjected to nonsense mediated decay, although currently we do not have any evidence for that. However, the majority of DAS genes did not show an overall change in expression, indicating that it is unlikely that the main mode of action of RiSP7 is to target mRNAs towards the NMD pathway. Among the alternative splice events observed in response to ectopic expression of *RiSP7*, most of them corresponded to intron retention (44%) followed by changes in the alternative 3' or 5' splicing site (Supplementary Data 4).

Given that one the main functions of SR45 is the modulation of alternative splicing, we hypothesized that *RiSP7* overexpression could affect the splicing of a subset of the targets of SR45 (genes able to be bound by SR45). Therefore, in order to get insights into the involvement of SR45 in the RiSP7-mediated alternative splicing, we obtained the putative orthologues in *Arabidopsis* of the DAS and DEGs potato genes regulated in response to RiSP7.

We then compared our data set of DAS genes with the SR45 associated RNAs (SARs) from two studies in *A. thaliana*[28,69], in which SR45 mRNA targets were identified by RNA immunoprecipitation (Figs. 7a–c). From 87 putative *Arabidopsis* orthologues of RiSP7-mediated potato DAS genes (Supplementary Data 4), 48 were

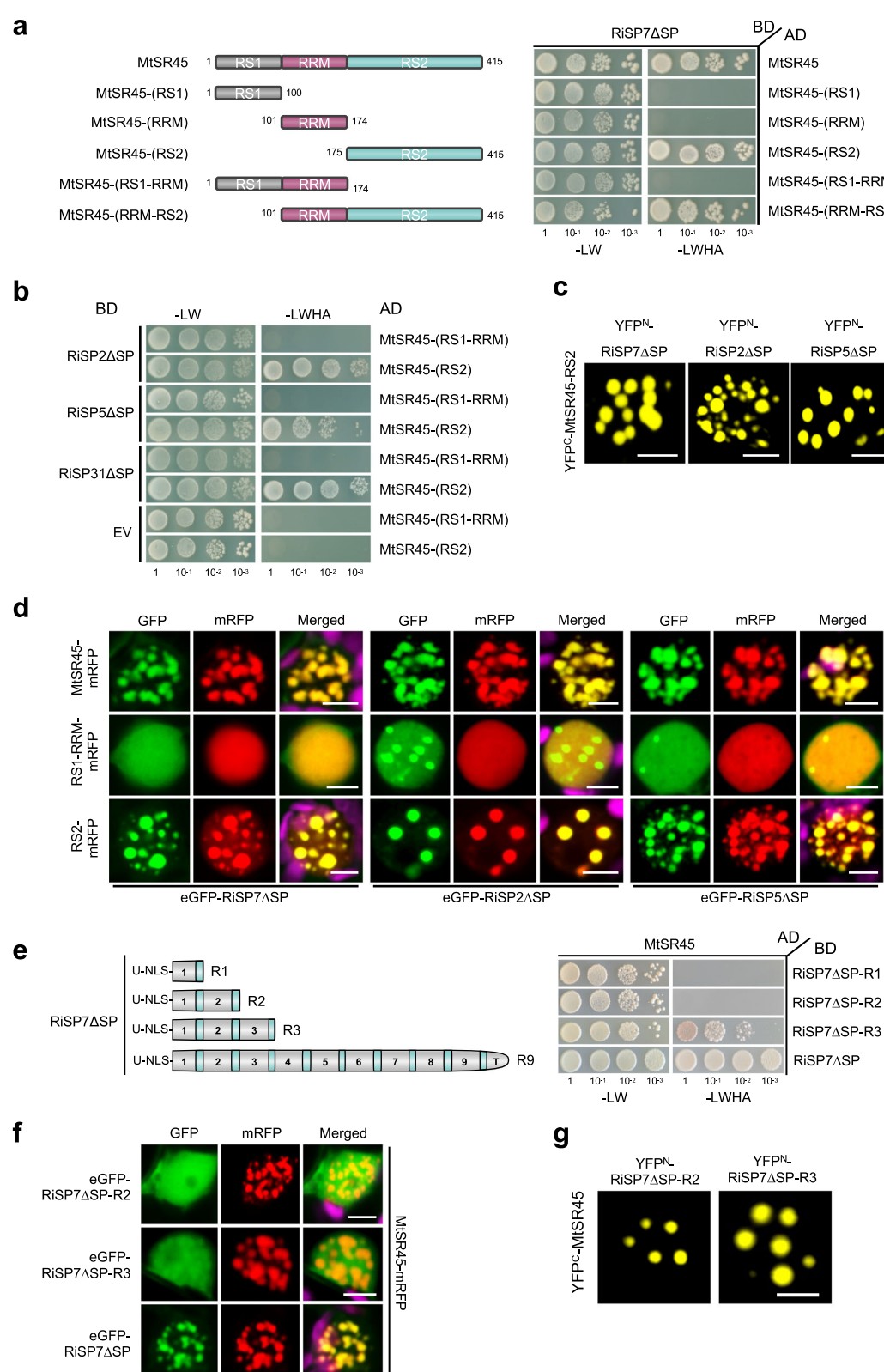

identified as AtSR45 mRNA targets (Figs. 7a, b and Supplementary Fig. 17). Among the differentially spliced genes in response to RiSP7 there are many transcriptional regulators (Fig. 7c), which could impact on the gene expression of downstream genes increasing the transcriptome complexity as suggested for the SR45 function in *Arabidopsis*[28]. However, the overlap between potato SP7-DEGs and SAR genes (Supplementary Fig. 17) and the AS analyzes of those genes in

potato showed that a large majority were not found differentially spliced. This suggests that their expression could be regulated by the SP7-SR45 complex by other mechanisms including chromatin remodeling as it has been suggested for SR45 and RNPS1[44,46,62] (Supplementary Fig. 17). Interestingly, the comparison between the orthologues in *Arabidopsis* of DEGs in response to SP7 to the DEGs in the *sr45* background shows an overlap of 11% genes with an enrichment

**Fig. 4 | The RS2 domain of MtSR45 is required and sufficient for effector interaction and number of repeats determine interaction strength. a–f** All Y2H spotting assays were performed using different yeast dilution series (1-10⁻³) with SP7-like effector constructs as prey (AD) and MtSR45 constructs as bait (BD). Positive interactions are indicated by colony growth on media lacking leucine, tryptophan, histidine, adenine (-LWHA). EV = Empty vector. For localizations and Bimolecular fluorescence complementation (BiFC) assays, single plant nuclei are shown after transient expression of fluorescent tagged proteins under control of the P35S promoter in *N. benthamiana* epidermal cells. Scale bars represent 5 μm. Experiments were performed with SP7-like effectors lacking their signal peptides (ΔSP). **d, f** Chloroplast autofluorescences are pseudo-colored in magenta (merged pictures). **a** Interaction assay of RiSP7 with different MtSR45 domains. Left side: Overview of generated MtSR45 truncation constructs. Right side: Y2H using RiSP7ΔSP against MtSR45 and truncations. Only proteins containing the RS2

domain interacted with RiSP7ΔSP. **b** Y2H as described in (a) with additional SP7-like effectors. **c** Confirmation of RS2 domain as effector target site using BiFC (RS2 domain fused to C-terminal half of YFP, YFPᶜ, SP7-like effectors fused to N-terminal half of YFP, YFPᴺ). **d** Co-localization of RiSP7ΔSP, RiSP2ΔSP or RiSP5ΔSP (N-terminal GFP fusion) with C-terminal mRFP fusions of MtSR45, RS1-RRM or RS2 domain alone. Only expression of RS2 domain containing proteins led to co-localizations into nuclear condensates. **e** Interaction assay of truncated RiSP7 versions. Left side: Overview of generated RiSP7ΔSP truncations (Full length = R9, 1–3 repeats = R1-R3). Right side: Y2H using RiSP7ΔSP truncations against MtSR45. **f** Co-localization of RiSP7ΔSP truncations (N-terminal GFP fusion) with MtSR45-mRFP. The minimum repeat number for effector re-localization into MtSR45 nuclear condensates was three. **g** BiFC of RiSP7ΔSP-R2 truncation (fused to N-terminal half of YFP, YFPᴺ) and MtSR45 (fused to C-terminal half of YFP, YFPᶜ) showed interaction in nuclear condensates.

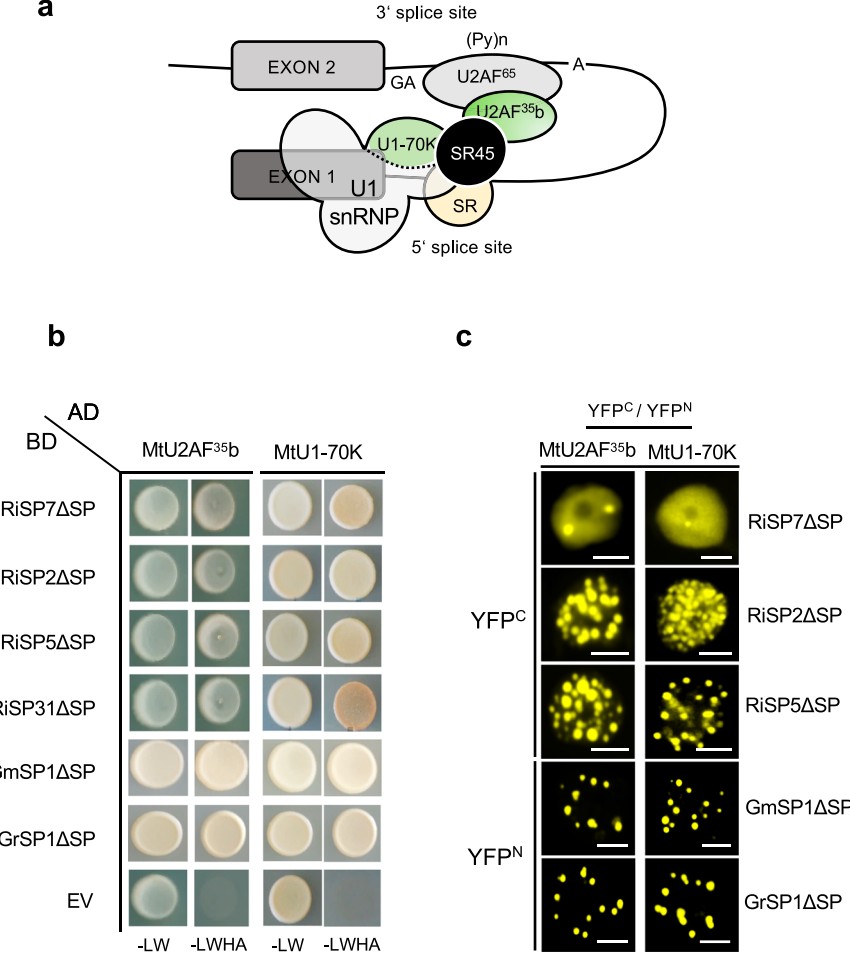

**Fig. 5 | SP7-like effectors interact with SR45 associated spliceosomal components U1-70K and U2AF³⁵b. a** Illustration of known interaction network between *Arabidopsis* SR45 and spliceosomal proteins at early stages of pre-mRNA splicing. AtSR45 directly interacts with U1-70k (part of U1 snRNP) and U2AF³⁵b (forms heterodimer with U2AF⁶⁵) at splice sites (Golovkin and Reddy, Ali et al., Day et al.). This interaction network may serve as bridging platform to facilitate splice-site pairing across introns and to recruit components for spliceosome complex formation (Day et al.). The dual interaction of SR45 with U1-70k and U2AF³⁵b is conserved in *Medicago* (see Supplementary Fig. 9). GA = 3′ splice site, (Py)n = Polypyrimidine tract, A = branchpoint adenosine, SR = additional SR proteins. **b** Direct Y2H assays showing that all SP7-like effectors (BD) interacted with *Medicago* U2AF³⁵b and U1-70k spliceosome components (as Preys, AD). Positive interactions are indicated by colony growth on media lacking leucine, tryptophan, histidine, adenine (-LWHA).

EV = Empty vector. **c** Validation of interactions seen in (a) by BiFC assays in *N. benthamiana* leaf epidermal cells. Shown are magnified pictures of single plant nuclei. Reconstituted YFP fluorescence was detected in nuclear condensates for plants co-expressing either MtU2AF³⁵b or MtU1-70k with all tested SP7-like effectors under control of the P35S promoter. For RiSP7ΔSP only a few nuclear condensates could be observed. Instead, YFP signals were mainly distributed evenly throughout the nucleoplasm. For MtU2AF³⁵b and MtU1-70k N-terminal fusions of the N-terminal half of YFP (YFPᴺ) were used for co-expression with N-terminal fusions of RiSP7ΔSP, RiSP2ΔSP and RiSP5ΔSP (fused to C-terminal half of YFP, YFPᶜ), while for experiments with *Gigaspora* members the YFP fusion halves (YFPᴺ, YFPᶜ) were interchanged. Scale bars represent 5 μm. All experiments were performed with SP7-like effectors lacking their signal peptides (ΔSP).

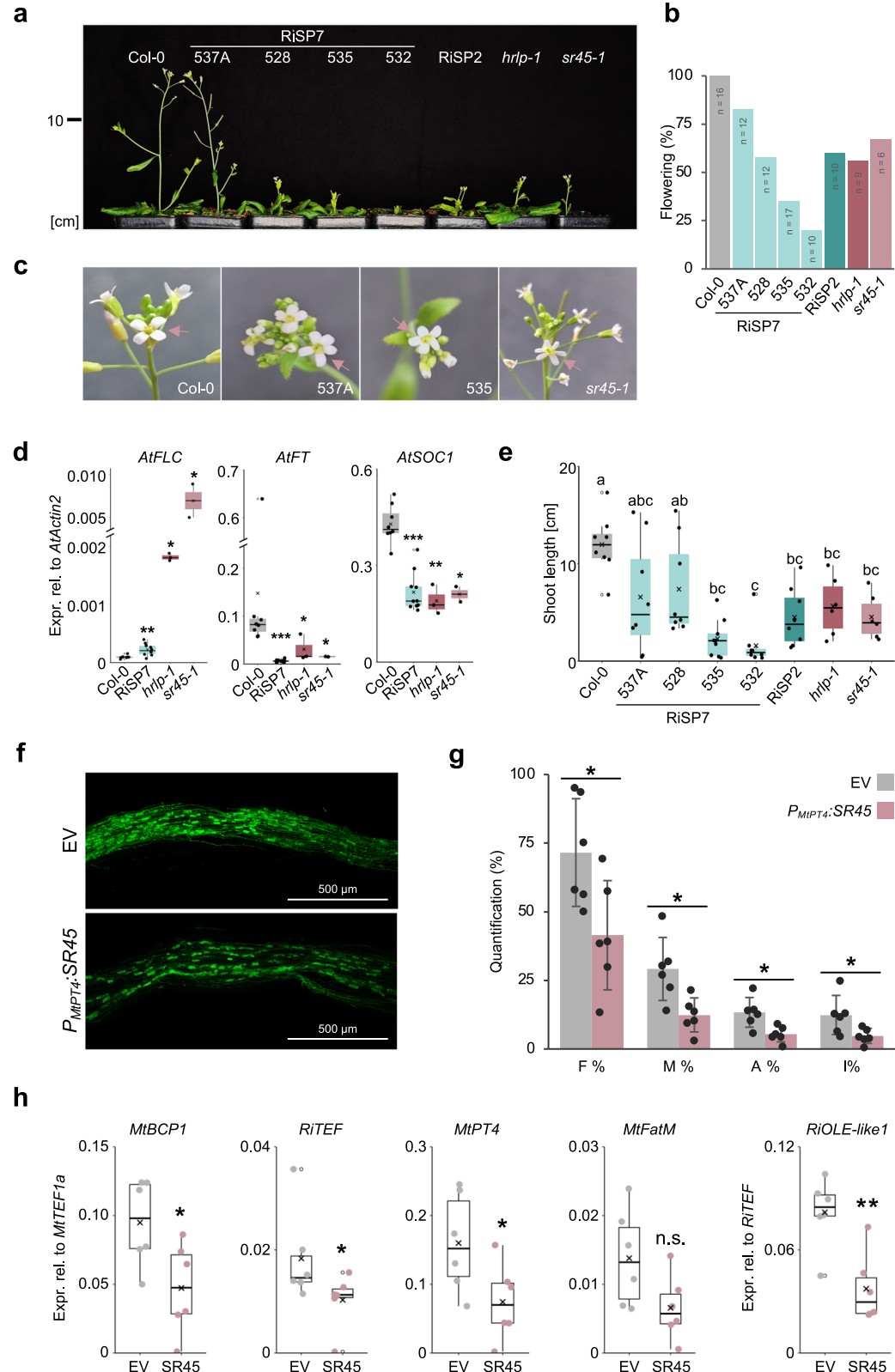

in immunity related genes as identified by GO term analysis (Supplementary Fig. 17 and Supplementary Data 4). This in line with previous studies where SR45 was shown to be a suppressor of innate immunity[28,34].

Among the genes differentially spliced in response to RiSP7 and whose orthologues in *Arabidopsis* were SR45 associated (48), we randomly selected 11 genes showing different types of AS events. We

analyzed the transcript levels of their different splicing variants to confirm their differential expression in response to RiSP7 using qRT-PCR (Fig. 7d). From those genes, most of them (6) had splicing events that altered the untranslated regions (UTR) or produced premature stop codons (4), while only one gave rise to an alteration of the coding sequence without affecting the stop codon. We confirmed that 9 of those genes underwent alternative splicing in response to RiSP7.

**Fig. 6 | Ectopic expression of SP7-like effectors in *A. thaliana* has a negative effect on the function of SR45, a putative negative regulator of the AM symbiosis. a** Growth phenotype of representative independent transgenic *Arabidopsis* plants grown for 32 days ectopically expressing *RiSP7* (lines 537 A, 535, 532, 528) or *RiSP2*. Promoters: Line 537 A - LjPUbi; Lines 535, 532, 528 - 2XP35S, *RiSP2* – P35S. **b** Percentage quantification of plants showing at least one fully developed flower after 32 days. Numbers represent total amount of plants from two independent experiments. **c** Example pictures of flowers with abnormal petal numbers (≥ 5; arrows) occasionally observed for transgenic *RiSP7* lines and *sr45-1*, while the WT usually forms 4 petals. **d** Expression levels of *AtFLC*, *AtFT* and *AtSOC1* relative to *AtActin2* in *Arabidopsis* expressing *RiSP7* or *sr45-1* and *hrlp-1* mutants compared to WT. **e** Growth measurements of stems during bolting. SP7-like effector expressing plants, hrlp-1 and sr45-1 mutants show a significant reduction in stem elongation compared to the WT. **f** Representative pictures of mycorrhized *M. truncatula* roots

transformed with an empty vector (EV) or with *MtSR45* expressed under the *MtPT4* promoter. Scalebars represent 500 μm **g** Quantification of root colonization as described by Trouvelot et al. expressed as mean ± SD. Quantified parameters: F% (frequency of mycorrhization), M% (intensity of colonization), A% (arbuscule abundance) and I% (hyphae abundance). **h** Relative expression of AM marker genes showing the quality and state of the mycorrhization. In all boxplots, the boxes show the quartiles, the whiskers mark the maximum and minimum values, except outliers that are shown as extra dots. Statistical significance: For (d, g and h) significance was calculated using T-test or Mann-Whitney U test (see "Method" section). For (**e**) one-way ANOVA and Tukey HSD post-hoc test were performed. Sample sizes: For (**b**) n = 8 (WT); n = 10 (RiSP7 lines: 528, 532 and 535); n = 3 (*hrlp-1*); and n = 2 (*sr45-1*). For (**e**) n = 8 (RiSP7 lines 537 A, 528, 532, RiSP2 and WT); n = 9 (RiSP7 line 535); n = 6 (*hrlp-1* and *sr45-1*). For (**g**) and (**h**) n = 6. Significances shown with letters or asterisks. Exact *p*-values are shown in Source Data.

Towards evaluating whether these alternative splicing events were also induced by other SP7-like effectors, transgenic potato plants expressing *RiSP5* were generated. Strikingly, 6 from the 9 DAS confirmed genes in *RiSP7* transgenic potato were also alternatively spliced in response to the ectopic expression of the *RiSP5* fungal effector (Supplementary Fig. 19). Furthermore, in *A. thaliana*, in which only 8 from the 11 genes analyzed in potato had more than one alternative splice form, seven of them were also alternatively spliced in response to RiSP7 and, most interestingly, 5 genes also underwent AS in response to SR45 loss of function (Supplementary Fig. 20). Taken together, our results suggest a conserved mechanism and a splicing target overlap for this effector family in different plants.

## Discussion

Plants are constantly challenged by interacting microbes and adapt their physiology modulating their gene expression accordingly. In recent years it has become evident that part of the plant physiological response to microbes also involves alternative splicing as a post-transcriptional mechanism to allow fast adaptation to the environment[26]. Although the mechanisms are starting to be elucidated, it has been shown that in several pathogenic interactions, these changes in AS can be directly caused by the microbe through the delivery of effector proteins, mainly with the aim to modulate immunity. Here we show that symbiotic arbuscular mycorrhizal fungi have evolved an effector family, the SP7-like family, able to interact with the plant mRNA processing machinery and induce AS of a small subset of genes. The founder member of this effector family, SP7, was identified in *R. irregularis* using a genetic screen for secreted proteins that contained an NLS in their sequence[9]. There, we could already identify other paralogues having a similar protein structure (hydrophylic tandem repeats) but only SP7 was functionally analyzed and shown to localize in the nucleus. The release of the first *R. irregularis* genome revealed several other putative paralogues[10,11], but only now, with the publication of a largely improved sequence and annotation[12,13], the full SP7-family could be disclosed. Our phylogenetic analysis shows that the SP7-like family seems to be unique to the Glomeromycotina and widespread among different orders. However, the fact that we could not find evidence of this family in the orders Paraglomerales or Archaeosporales suggests that the ancestor of this gene family might not have been present in the earliest AM fungi[70]. Most SP7-like genes are located in the B compartment of the *R. irregularis* genomes which is transcriptionally repressed by methylation in axenic culture[12]. The high copy number of SP7-like effectors in the *R. irregularis* genomes could be due to the less evolutionarily constrained situation of compartment B which is less enriched in core genes[12]. Interestingly, secreted proteins including effectors in this compartment were shown to be de-repressed during the *in planta* colonization[12]. Heterokaryotic strains showed a general loss of some of the SP7-like effector copies. There is not much know about the difference in symbiotic behavior of homo-

vs. heterokaryotic strains, but it has been recently shown that heterokaryotic strains are less efficient at promoting plant growth[71]. It would be interesting to know if SP7-like effectors could be involved in symbiotic performance.

We had previously shown that RiSP7 localizes at the nucleus, and that its ectopic expression in roots is able to promote biotrophy and counteract defense responses[9]. The identification of novel members in the SP7-like family lacking a predicted NLS challenged our hypothesis that these proteins would be nuclear effectors modulating immunity. However, localization assays *in planta* of different members of the family revealed that they all targeted the nucleus. This might be explained by the presence of the basic amino acids KR in the conserved signature present in each repeat (Fig. 1) or other non-conventional NLS[19]. The amino acid analysis of SP7-like effectors revealed that their composition is biased towards the presence of five specific amino acids (Asp, Tyr, Lys, Ser and Gly) that represents between 75 and 80% of their sequence (Supplementary Fig. 21). This is in agreement with their nature as proteins containing large intrinsically disordered regions (IDRs) which are characterized as having low sequence complexity and being enriched in a limited number of amino acids such as those in SP7-like proteins[56]. Proteins with large IDRs usually lack a defined 3D structure but are often organized in repeat elements, what allows multivalent intermolecular interactions. They are common components of biomolecular condensates, particularly those in which RNA accumulates such as nuclear condensates, P bodies or stress granules[56]. This localization is due to the low amino acid diversity that promotes the appearance of blocks of positively and negatively charged amino acids that are critical for targeting to RNA granules. This is consistent with the observed localization of SP7-like proteins in nuclear condensates and also in P bodies.

Biomolecular condensates can mediate specific reactions or increase the dwell time by bringing molecules together or serve the sequestration of specific proteins in plants[72], where they are increasingly recognized as playing a role in immunity[73]. Thus, the localization of SP7-like proteins and their interactome supports a role for these effectors in mRNA processing. We speculate that they are able to do so by bringing together specific mRNAs and RNA processing proteins. This is also consistent with the ability of proteins having IDRs to drive multivalency, the ability to interact with multiple partners through several interaction sites, forming a highly connected network through noncovalent crosslinks with other proteins[56,57]. These interactions might be weak individually what allows for liquid-liquid phase transitions[56,57] as observed when analyzing the strength of the interaction between single or multiple repeats of RiSP7 and MtSR45.

The fate mRNAs is strongly determined by the set of RNA binding proteins that accompany the transcript from early stages of mRNA biogenesis until their translation, from which SR proteins play a major role[37,38]. Our interactome results show that SP7-like effectors appear associated with many components of this pathway, and most

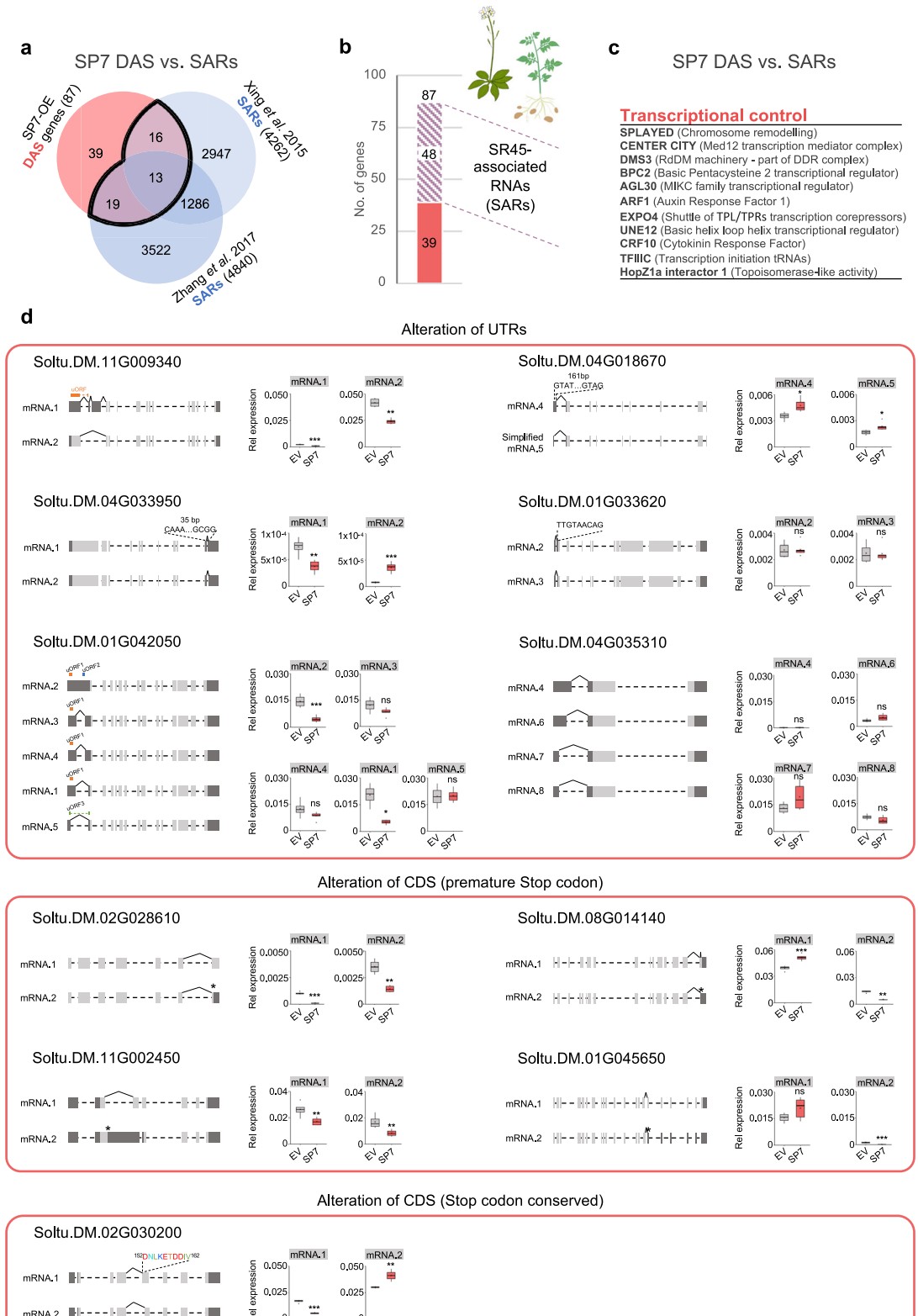

prominently with the unconventional SR protein SR45 and the splicing factors U2AF35b and U1-70K. Taken together, this suggests that impacting on alternative splicing might be a key function of this effector family. However, other identified components suggest that these effectors might not be involved only in early stages of mRNA processing but also accompany the mRNAs and SR45 through their journey to translation. This is not totally surprising since SR proteins

have sometimes been found to remain associated to specific mRNAs and participate in their export out of the nucleus and in their translation[37]. In fact, most SR proteins shuttle between the nucleus and the cytoplasm, with some of them being export adapters for specific mRNAs out of the nucleus, such as SRSF3 and SRSF7[74]. RSZ22, the orthologue of SRSF7, and involved in immunity[75], was retrieved as interactor of two SP7-like proteins (Supplementary Data 3, and

**Fig. 7 | Extopic expression of SP7 alters the plant's alternative splicing landscape. a** Venn diagram showing the overlaps of *Arabidopsis* SR45 associated RNAs (SARs, blue) identified by Xing et al. and Zhang et al. in comparison with the *Arabidopsis* orthologs of DAS genes (red) identified in SP7 expressing (2XP35S promoter) potato plants. **b** Stacked bar chart from comparisons shown in (a) illustrating that 48 out of the 87 SP7 induced DAS genes (*Arabidopsis* orthologs of potato DAS) are also known *Arabidopsis* SR45 RNA targets. **c** List of SP7 induced DAS genes that function in transcriptional control. **d** Splicing models and quantification of DAS gene isoform expression levels using qRT-PCR. From the 48 SP7 regulated DAS genes that are known SARs (**a**–**b**), 11 genes were randomly selected to measure the relative expression of each splicing isoform. Splicing models:

dashed lines represent introns, while boxes indicate exons. Boxes in dark gray correspond to untranslated regions (UTRs) and light gray to protein coding sequence (CDS). Expression values relative to *StActin* are shown as boxplots in which the boxes show the quartiles and the whiskers mark the maximum and minimum values, except outliers that are shown as extra dots. Statistical significance was assessed by performing a Student T test or a Mann Whitney U test after checking the normal distribution of the data and homocedasticity as explained in Materials and Methods. Sample size was $n = 4$ for EV (from two independent lines) and $n = 6$ for SP7 (from three independent lines). Significance levels: n.s. (non-significant) $p > 0.05$, * $p < 0.05$, ** $p < 0.01$, *** $p < 0.001$. Exact $p$-values are shown in Source Data.

Supplementary Figs. 9–10). Furthermore, other proteins controlling the nuclear-cytoplasmic shuttling such as the chaperons HSP90, 70, 40 and 60[76], the specific SR protein kinases such as SRPK1[76], the helicase DRH1 required during defense responses[77], or ALY1, a component of the TREX complex[78] were also identified in the SP7-like interactome. We could show that SP7-like effectors directly interact with ALY1 in nuclear bodies, further supporting our model that these effectors accompany mRNAs in their fate from transcription to translation (Supplementary Figs. 9, 10 and 12). And even more, several ALBA proteins known to bind specific mRNAs and control their translation by phase separation between stress granules and P bodies[79] were identified in the interactome of SP7-like effector. Altogether, we postulate that SP7-like effectors are delivered to the nucleus to alter the fate of specific mRNAs by interacting with the plant mRNA machinery to alter their splicing, transcription and/or their translation.

In this context, the main interactor of SP7-like effectors, the SR45 protein is a crucial component. SR45 is the orthologue of the well-studied human RNPS1 protein that plays a dual role in splicing and mRNA quality control[41,42]. Furthermore, as a component of the ASAP complex they have also been proposed to participate in transcriptional regulation by chromatin remodeling[43,46]. RNA immunoprecipitation assays with SR45 in *Arabidopsis* have shown that more than 4000 gene transcripts are associated with this protein, including several from intron-less genes[69]. Given that SR45 has been involved in many developmental, stress and immunity related processes[28,39,47,49,50,80], explaining the pleiotropic phenotypes of the *sr45-1* mutant, it is clear that target specificity must be given by other RNA binding regulators. The fact that all SP7-like proteins investigated interacted directly also with the core splicing proteins U1-70K and U2AF35 suggest that their function begins at the onset of splicing possibly already during transcription.

Alternative splicing is increasingly recognized to play a key role in controlling innate immunity in plants, as many mutants in splicing components show an immunity-related phenotype[25]. For example, CPR5, an *Arabidopsis* RNA binding protein from the serine-arginine rich family that localizes in nuclear speckles, has been recently shown to modulate plant immunity. It does so by interacting at those speckles with two key regulators of RNA processing, the NineTeen Complex and the cleavage and polyadenylation specificity factor, to control the AS of more than 500 transcripts[30]. Interestingly, both of those components were found as interactors of SP7 (Supplementary Data 3). It is thus not surprising that AS has been shown to be a target for several effector proteins in order to subvert plant immunity[33,34]. Because most of the splicing events observed were not paralleled to changes in transcript levels of the genes affected, we assume that the production of proteins with different sequence or domain arrangements, or mRNAs with different regulatory regions, might be the main function of SP7-like proteins on these transcripts. This might serve different purposes, such as producing different subcellular localization, changes in protein stability, alteration of translation speed, or even producing dominant-negative versions of the target proteins.

There are several examples of microbial effectors that can target components of the host mRNA processing, but only a few of them have been shown to target host alternative splicing. One of them is HopU1 from the bacterial pathogen *Pseudomonas syringae*. This effector had been identified as an ADP-ribosylation factor of RNA binding proteins, GRP7 and GRP8, interfering with immunity[35,81]. GRP7 can alter the choice of alternative 5' splice sites of specific mRNAs[82]. Further studies revealed that in the case of GRP7, HopU1 impaired immunity by preventing the interaction of GRP7 with the mRNA of immune receptors such as FLS2 and EFR and decreased their protein abundance. However, it was not clear if this was achieved by an alternative splicing mechanism[83]. In contrast, several other studies have directly involved effectors in the modulating AS in their plant hosts[36,68,84,85] Thus, GRP7 is also the target of the *Phytophthora sojae* effector FYVE1 that decreases its binding affinity for the RZ-1A protein impacting on the AS and transcription of immunity related genes[85]. MiEFF18 is an effector from *Meloidogyne incognita* that interact with the spliceosomal protein SmD1 and modifies AS impacting on giant cell formation[36]. Interestingly, the orthologue of SmD1 was also immunoprecipitated by SP7 (Supplementary Data 3). Also, two seminal studies from Huang and collaborators[33,34] revealed that alternative splicing by effectors is an important mechanism to reprogram plant immunity in the oomycete pathogen *Phytophthora* (from 87 RXLR effectors, 9 regulate splicing). One of them, PsAvr3c, interacts with SKRP-like proteins and alter alternative RNA splicing of more than 400 genes. SKRP proteins turned out to be negative regulators of plant immunity, but most interestingly immunoprecipitation assays showed an interaction with SR45 and with U1-70K and U2AF35 suggesting that they are part of the spliceosome. They could show that both PsAvr3c and SKRP ectopic expressions induced alternative splicing and that SKRP targets exon 3' end of unspliced RNA[34,80]. This reveals a similar picture to the function of SP7-like effectors and their interactors, although the amount of alternative spliced genes in response to ectopic expression of the mycorrhizal effector is much lower (20 times). This is also not surprising, because in contrast to the large amount of alternative spliced plant genes observed in response to Phytophthora infection, more than 5000 genes in tomato[33], the number of alternative spliced genes during arbuscular mycorrhiza colonization was shown to be very low, with only 500 events in pea roots[86]. Although we believe that certainly the number of AS during mycorrhizal symbiosis is lower than in the case of *Phytophthora*, we think that this number is somewhat falsified due to the dilution effect. We expect most AM fungal effectors *in planta* to be expressed and secreted at the major symbiotic interface, the arbuscule[17]. Thus, the analysis of whole roots might obscure the real number of AS transcripts. Furthermore, root colonization by AM fungi is not a synchronous process, and therefore, analyzing whole roots might also not show the dynamic variation in AS during the colonization process.

Our results here demonstrate that the family of SP7-like effectors from AM fungi localize to nuclear condensates and interact with components of the plant mRNA processing pathway at those loci. We propose that SP7 like proteins act as negative regulators of the splicing

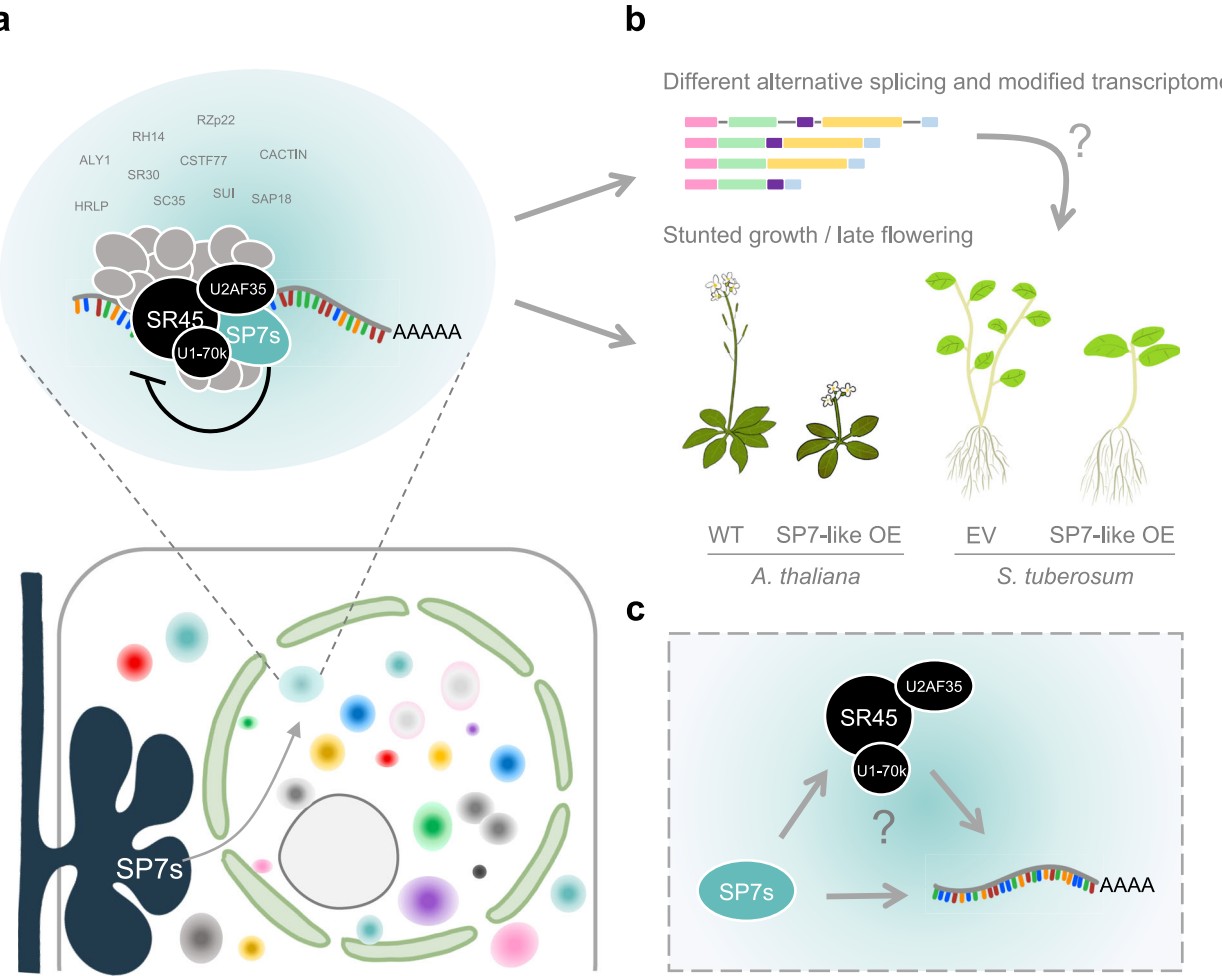

**Fig. 8 | Model for the mode of action of the SP7 effector family in plants. a** SP7-like effectors (SP7s) are translocated in the plant nucleus in arbuscule-containing cells. They localize to discrete plant nuclear condensates where they interact with SR45 and additional plant mRNA processing proteins. These interactions have a negative impact on the activity of SR45 and other splicing factors. **b** *In planta* expression of SP7-like effectors results in stunted growth and late flowering phenotypes characteristic for *sr45-1* and other splicing mutants. This is accompanied by changes in the splicing pattern and a modified transcriptome of specific SR45-associated gene subsets. **c** An open question is how the transcript target specificity is achieved. One possibility is that it is encoded in the sequence of the effectors themselves, allowing them to be recruited to specific mRNAs. But also, interaction of effectors with specific SR proteins could determine target specificity. Future RNA immunoprecipitation assays will help to clarify this and other open questions.

protein SR45 to modulate the AS program of their host plants and/or the fate of specific mRNAs in arbuscule-containing cells (Fig. 8). As a result of their negative regulation of SR45, the systemic expression of SP7 in plants (potato and Arabidopsis) results in a stunted and late flowering phenotype. Among the potato DAS and DEGs genes in response to RiSP7 there are some candidates that could be involved in reprogramming the immune system of the plant and prevent defense reactions. However, we do not exclude that other plant developmental or nutritional processes could be targeted by this effector family that would indirectly impact on the immune system of the plant to promote symbiotic biotrophy as we had previously proposed[9]. Understanding the mechanisms of communication between mutualistic symbionts and their crop plants will help to identify targets that can be engineered for mechanisms of resilience and improved nutrient acquisition.

## Methods
### Bioinformatic online resources used in this study
Signal peptide prediction for SP7-like effector proteins was carried out using SignalP5.0, (https://services.healthtech.dtu.dk/services/ SignalP-5.0/)[87]. The presence of possible NLS in SP7-like effectors was predicted using the traditional PSORTII prediction of WoLFP-SORT (https://wolfpsort.hgc.jp/)[88] with plant as the selected organism type. Disordered regions in SP7-like effectors and MtSR45 were predicted using PrDOS, (http://prdos.hgc.jp/cgi-bin/top.cgi)[89] and IUPred3 (https://iupred3.elte.hu/plot)[90] with default parameters. Alignments and comparative percentage identity analysis were performed using Clustal Omega (http://www.ebi.ac.uk/Tools/msa/ clustalo/)[91]. For analysis of consensus motifs, highly conserved regions of always the second repeat from all 21 identified full length SP7-like effectors (Supplementary Data 2, Supplementary Fig. 1a) were manually aligned and visualized with ESPpript 3 (http://esript. ibcp.fr/ESPript/ESPript/)[92]. Aligned repeat regions were further used in WebLogo (http://weblogo.berkeley.edu)[93], to create the consensus logo with the effector family characteristic Y[KR]Rx[AP]x[AP] motif. Protein, gene and *R. irregularis* chromosome models were scaled using IBS[94] and redrawn according to these scales. The evolutionary relationship of SP7-like effectors was inferred by using the Maximum Likelihood method and JTT matrix-based model using MEGA11[95] with 1000 bootstrap replicates. For that purpose, the protein sequences

of all 21 identified full length SP7-like effector members were first aligned using ClustalW in MEGA11. Initial trees for the heuristic search were obtained automatically by applying Neighbor-Joining and BioNJ algorithms to a matrix of pairwise distances estimated using the JTT model, and then selecting the topology with superior log likelihood value. The tree with the highest log likelihood is shown (Fig. 1c) and visualized using iTOL (https://itol.embl.de/)[96]. Bootstrap values are indicated by node circles. Branch lengths are ignored. The phylogenetic tree of SR protein members was inferred using MEGA7, rooted Neighbor-Joining method, JTT modeling and Bootstrap with 1000 replicates. Evolutionary distances were computed using the p-distance method) after ClustalO alignment. Bootstrap values and branch lengths are ignored. To identify SP7-like genes in Mucoromycota species, genome data for Glomeromycotina, 19 different Mucoromycotina and Mortierellamycotina species as well as genomes of *Laccaria bicolor*, *Tuber melanosporum*, *Serendipita indica* and *Serendipita vermifera* were BLAST searched using BlasterQt[97] to create a local BLASTp interface. Protein sequences and motifs of known[9–11] SP7-like effectors were used as query. BLASTp hits and respective gene models thereof were further analyzed for the existence of predicted signal peptides, repeats, conserved motifs and introns. Identified SP7-like effector protein and gene sequences are summarized in Supplementary Data 2. To assess whether SP7-like effector transcripts are present in AM species and different fungi outside the Glomeromycotina, effector coding sequences were used as query to perform BlastN analysis in SRA (Sequence Reads Archive) databases. For that purpose, publicly available RNA based experiments (Filter: RNA) from organisms belonging to Mucoromycotina, Blastocladiomycota, Zoopagomycota, Mortierellomycotina and Glomeromycotina were separately selected and used for Blast analysis with the parameter "more dissimilar sequences". For Chytridiomycota (*Batrachochytrium dendrobatidis*), Ascomycota (*Blumeria graminis*, *Laccaria bicolor*), Basidiomycota (*Uromyces spp.*, *Serendipita vermifera*) and *R. irregularis* exemplary experiments were selected. All obtained reads are listed and compared in Supplementary Data 1. In all cases, the criteria used to assign reads as significant matches were a) they would show at least 75% identity within the alignment; b) an alignment length should be over 94 bp; and c) at least one of these reads spans the SP7-like family consensus motif after translation. GO (Gene ontology) term enrichment analyses of identified plant interaction partners were performed using g:GOSt functional profiling at the g:Profiler web service[98].

To confirm the genomic location and variation of SP-like genes in homo- and heterokaryotic strains of *R. irregularis* we carried out BLASTN-based sequence homology searches using the nucleotide sequences of SP-like genes in Supplementary Data 2 as query sequences. We then filtered out partial hits or potential pseudogenes by only retaining sequences that pass a ≥ 90% identity and > 50% query coverage threshold and contain a start codon. This analysis confirmed our TBLASTN-based results (Fig. 1) with the exception that we identified additional truncated copy of RiSP1-3 on contig CP110690.1 with ~54% query coverage and ~95% sequence identity. We then plotted the location of all identified SP-like genes using the KaryoploteR package in R[99]. In addition, we performed pairwise whole genome alignments comparing the assembly of isolate DAOM-197198[13] to available homo- and heterokaryotic strains[12,51] using the nucmer program of the MUMmer package (v4)[100]. We then selected for alignments >10 kB using the delta-filter function of nucmer and plotted the results using mummerplot and gnuplot. Finally, to visualize presence/absence variation or translocations of SP-like genes we added the coordinates of SP-like genes and prepared the final figures using Affinity designer 2.

Gene lists of 918 DEGs in response to the loss-of-function of SR45 and 4262 SARs were retrieved from the work of Xing et al., while 1791 DEGs (identified by the TopHat2 pipeline) and 4840 SARs were obtained from Zhang et al. These lists were compared to the lists of *Arabidopsis* putative orthologs of the potato DEGs and DAS genes in response to the ectopic expression of *RiSP7*. The orthology search for the DEGs done by selecting the best reciprocal BLASTp hit. For GO term enrichment analysis of the SP7-OE potato DEGs, the respective *A. thaliana* orthologs were used at ThaleMine v5.1.0.

## Biological material and transformation of plants

*Rhizophagus irregularis* DAOM 197198 was cultivated in monaxenic culture as described before[101]. *Gigaspora margarita* (BEG34) spore inoculum was obtained from Dr. J. Palenzuela (EEZ, CSIC, Spain) and *Gigaspora rosea* (DAOM194757) spore inocula were donated by Dr. C. Roux (CNRS, Toulouse, France) and from Dr. S. Roy (Agronutrition, France).

Transgenic *A. thaliana* plants were generated using the floral dip method[102] with Agrobacteria (strain GV3101) containing the desired expression constructs. Transgenic plants were further selected (kanamycin, DsRed fluorescence marker) and propagated for homozygous seeds. Seeds of *A. thaliana sr45-1* (SALK_004132) and *hrlp-1* (SALK_124411C) mutants in the Col-0 background were purchased from the ABRC Arabidopsis biological resource center.

*Medicago truncatula* cv. Jemalong A17 seedlings were transformed with *Agrobacterium rhizogenes* strain ARqua1 and generated composite plants were mycorrhized and cultivated as previously described in ref.103 with the following modification: Fertilization of plants was carried out with 5 ml of half strength Long Ashton nutrient solution following a decreasing phosphate regime every week: 20, 10, 10, 0 and 0 μM). Plant roots were harvested for quantification and microscopic analysis 5 weeks after R. irregularis (DAOM 197198) inoculation.

Leaf disk transformation of potato (*Solanum tuberosum*, cv. Desiree) was carried out using *Agrobacterium tumefaciens* (AGL-1 strain) with overexpression constructs for RiSP7, RiSP5 or an empty vector (EV). Transgenic plants were grown in a sand-gravel mixture (4:1) at 21 °C.

*N. benthamiana* plants were grown in soil at 28 °C. All plants had a light cycle of 16 h day/8 h night and were grown in growth chambers or a GroBanks Birghtboy-XXL.2 (CLF Plant Climatics, Wertingen, Germany).

## Genomic DNA isolation from fungal spores

Spores of *R. irregularis* DAOM 197198 (approximately 1000), *G. margarita* (BEG 34) or *G. rosea* (DAOM194757) (for both approximately 300 spores) were surface sterilized for 30 min (4% Chloramine T) and incubated in a 100 μg ml$^{-1}$ gentamycin and 200 μg ml$^{-1}$ streptomycin antibiotic solution. After washing with sterile $_d$H$_2$O spores were crushed in 0.1% sarkosyl in TE-buffer solution using a pestle for eppendorf tubes. Pronase (0.25 mg ml$^{-1}$) and 1% SDS were added to the spore solution and incubated at 65 °C for 1 h. Samples were homogenized every 15 min. Subsequently 1 M NaCl was added and solutions were incubated on ice for 30 min. Genomic DNA was then extracted and precipitated using phenol/chloroform and standard protocols.

## RNA isolation from plant and fungal material

For mRNA extraction frozen plant or fungal material was homogenized with 5 mm metal beads in a cell mill MM200 Retsch. The frozen material was homogenized in three cycles of 1 min at a frequency of 25/s, and refrozen in liquid nitrogen between each cycle. Plant and fungal mRNA was extracted using the innuPREP Plant RNA kit (Analytik Jena, Jena, Germany) or the TRIzol method (ThermoFischer Scientific, Waltham, MA, USA) according to manufacturer instructions.

## Molecular cloning and vector plasmid construction

The genomic or cDNA sequences of SP7-like effectors *RiSP7*, *RiSP2*, *RiSP5*, *RiSP31*, *GmSP1*, *GrSP1* were amplified and cloned in pCR2.1/TOPO, pCR8/GW/TOPO or pENTR/D-TOPO (ThermoFischer

Scientific, Waltham, MA, USA) with the primers listed in Supplementary Data 5. *MtSR45* (*Medtr2g069490*), *MtU1-70K* (*Medtr8g077840*), *MtU2AF35b* (*Medtr1g035130*), *NbSR45* (*Nb101Scf00283g0008*), *AtSR45* (*At1g16610*), *AtDCP2* (*AT5G13570*), *MtSR45A* (*Medtr3g462330*), two isoforms (*AtSR45A.2* and *AtSR45A.2b*) of *AtSR45A* (*At1g07350*), *MtSR30* (*Medtr1g083400*), two isoforms (*AtSR30.1* and *AtSR30.2*) of *AtSR30* (*At1g09140*) and *MtAly1* (*Medtr4g063557*) were amplified from cDNA of wild type plants and cloned in the same vectors.

For overexpression in potato and *A. thaliana* (lines RiSP7 528, 532 and 535) full-length cDNA sequences of SP7-like genes (including the UTRs and the protein signal peptides) were cloned into the Gateway destination vector 2xP35S-pKGWRedRoot[103]. For RiSP7 line 537 A in *A. thaliana* the vector pUB-GW-Hyg (LegumeBase accession No.: AB303065) containing the constitutive *Lotus japonicus UBI1* promoter was used. For that purpose, a DsRED and spectinomycin resistance cassette was inserted (pUB-GW-Hyg-RR). In addition, transgenic *A. thaliana* plants expressing RiSP2-GFP (cDNA sequence lacking its signal peptide) cloned in pCGFP-RR[101] were generated. For arbuscule specific overexpression of *MtSR45* in *M. truncatula* composite plants, the coding sequence was cloned into the pMtPT4-GW-Hyg-RR that was obtained by replacing the *LjUBI1* promoter from pUB-GW-Hyg-RR with the *M. truncatula* PT4 promoter[103].

For yeast two hybrid assays the coding sequences of SP7-like proteins lacking the signal peptide (*RiSP7ΔSP, RiSP2ΔSP, RiSP5ΔSP, RiSP31ΔSP*, and *GmSP1ΔSP* and *GrSP1ΔSP*) and from *MtSR45/AtSR45/NbSR45, MtU1-70K, MtU2AF35b, MtSR45A, AtSR45A.2a, AtSR45A.2b, MtSR30, AtSR30.1* and *AtSR30.2* as well as and RiSP7 or MtSR45 truncation constructs were amplified with restriction sites and subsequently cloned into the pGBKT7 or pGADT7-REC vectors (Takara Clontech Bio Europe, Saint-Germain-en-Laye, France). The prey vector constructs containing MtSR45 and MtU2AF35b were previously reported[9,16]. For localization or co-lozalization assays in *N. benthamiana*, the coding sequences of *RiSP7ΔSP, RiSP2ΔSP, RiSP5ΔSP, GmSP1ΔSP, GrSP1ΔSP, MtSR45, AtSR45, NbSR45, MtU2AF35b, MtU1-70K, AtDCP2, MtSR30* and *RiSP7* or *MtSR45* truncation constructs were cloned into the Gateway destination vectors pCGFP-RR, pNGFP-RR, pK7WGF2 or pK7RWG2[101,104,105]. The MtSR45-mRFP and RiCRN1-C-eGFP co-localization constructs were previously reported[16]. As co-localization control, free eGFP with stop codon was cloned into the vector pK7FWG2[104]. For *in planta* pulldown assays the coding sequences of *RiSP7ΔSP* and *RiSP7 +* SP were cloned in pCGFP-RR[101]. *RiSP7ΔSP, RiSP2ΔSP, RiSP5ΔSP, GmSP1ΔSP, GrSP1ΔSP, MtU2AF35, MtU1-70K, MtSR45A, MtAly1* and *RiSP7* repeat truncations as well as *MtSR45* full length or truncated cDNA versions were cloned into Gateway BiFC vectors P35S-pSPYNE-GW or P35S-pSPYCE-GW[106] for *in planta* interaction assays.

## RNA-seq and splicing analysis

Root RNA from 6 potato plants overexpressing RiSP7 (Two biological replicates of three independent lines) and 4 empty vector control (EV) plants (Two biological replicates of 2 independent lines) was isolated following the TRIzol method. The DNaseI-treated RNA was sent to BGI (Yantian District, Shenzhen 518083, China, https://www.bgi.com/global) for sequencing on a DNBseq platform (DNA nanoballs paired-end (100 bp) mode based DNBSEQ Technology). An Agilent 2100 Bio analyzer (Agilent RNA 6000 Nano Kit) (Agilent Technologies Inc., United States) was used for RNA quality control. For library preparation, mRNA enrichment was realized using oligo dT beads. RNA fragmentation and first-strand cDNAs were generated using random N6-primed reverse transcription, followed by second-strand cDNA synthesis. The synthesized cDNA was subjected to end-repair, 3'-adenylation and adapter ligation. The purified cDNA was enriched in several rounds of PCR amplifications. After sequencing, raw reads mapped to rRNA as well as low-quality reads, adapter containing reads

and reads with high content of unknown bases were removed using the BGI-internal software SOAPnuke (version 1.5.2). Clean reads were further mapped to the reference *Solanum tuberosum* v6.1[107] genome from Phytozome using HISAT2 v2.0.4[108] with following parameters: --phred33 --sensitive --no-discordant --no-mixed -I 1 -X 1000 --rna-strandness RF. After mapping, StringTie[109] was used to reconstruct transcripts and novel transcripts were predicted by using Cuffcompare[110]. The coding ability of those new transcripts was assessed using CPC[111]. Novel transcripts were merged with reference transcripts using Bowtie2 v2.2.5[112] to get a complete reference for downstream gene expression analysis. Gene expression levels for each sample were calculated with RSEM v1.2.12[113] using default parameters and based on those the differentially expressed genes (DEGs) were detected with the DEseq2 algorithms, performed as described[114]. Genes were considered as regulated with|log2FoldChange| >1 and *pValue* < 0.05 as thresholds. For the splicing analysis, after the genome mapping, rMATS v4.0.2[115] was used to detect genes that are differentially spliced between the groups of samples based on the relative abundances of the splicing isoforms of a gene across samples with following parameters: -t paired --nthread 8 --tstat 4. It calculated the *pValue* and false discovery rate (FDR) for the difference in the isoform ratio of genes in the EV samples versus the SP7-overexpressing samples. Genes with FDR ≤ 0.05 were defined as differential alternative splicing (DAS) genes (Supplementary Data 4).

## Gene expression analyzes

Relative expression values for the specific mRNA isoforms alternatively spliced detected in Supplementary Data 4 were verified by qRT-PCR. To that end cDNA synthesis and quantitative real time PCR (qPCR) were carried out as described[103]. The PCR protocol was selected as follows: 5 min 95 °C, 15 s 95 °C, 15 s Tm °C, 30 s 72 °C (40 cycles) on a CFX Connect Real-Time PCR Detection System (Bio-Rad Laboratories GmbH, Munich, Germany). The annealing temperature Tm was adjusted for each primer combination. Plant transcript levels were normalized to the levels of the corresponding housekeeping gene: *AtActin2* (At3g18780) or *AtActin8* (At1g49240) in *A. thaliana*, *MtTEF1α* (Medtr6g02180) in *M. truncatula* and *StActin* (Soltu.DM.05G024990) in potato, while fungal gene expression was normalized to the levels of *RiTEF1α* (DQ282611). Expression data are given as relative expression values calculated using the $2^{-\Delta Ct}$ method. Transcript levels of genes were determined in 3 independent lines ectopically expressing *RiSP7*, 1 line expressing *RiSP5* and 2 control lines expressing an empty vector (EV) in at least three biological replicates with two technical replicates per reaction. All primers are listed in Supplementary Data 5.

## Yeast two hybrid interaction assays

pGBKT7 bait vectors were transformed in *S. cerevisiae* AH109 (Takara Clontech Bio Europe, Saint-Germain-en-Laye, France) using the LiAc/SS carrier DNA/PEG Method[116]. Yeast-two-hybrid library screens were carried out according to Matchmaker® Gold Yeast Two-Hybrid System User Manual (Takara Clontech Bio Europe, PT4084-1) using a cDNA library of *M. truncatula* roots colonized with *R. irregularis* in *S. cerevisiae* strain Y187[9] or a normalized cDNA library of *A. thaliana* (Takara Clontech Bio Europe, Cat.# 630487). For direct interaction studies co-transformation of bait and pray was employed. Dilution series were spotted on yeast synthetic drop-out medium without leucine and tryptophan and on synthetic drop-out medium without leucine, tryptophan, histidine and adenine under sterile conditions and incubated for 4-5 days at 30 °C. Positive interactions are indicated by yeast colony growth on media lacking leucine, tryptophan, histidine and adenine.

## Transient protein expression in *N. benthamiana* leaves

Transient expression of proteins in *N. benthamiana* leaves was performed as described[106] using *Agrobacterium tumefaciens* strain GV3101 to infiltrate leaves of 4−5 weeks old plants. Final optical density

(OD$_{600}$) of *A. tumefaciens* was adjusted to 0.5 with AS medium supplemented with 2% sucrose (localization and co-localization studies) or to 1.0 (GFP pulldown assay and BiFC). To prevent silencing, an *A. tumefaciens* strain K5A018 carrying the p19 silencing suppressor of tomato bushy stunt virus[117] was co-infiltrated in a 1:1 ratio for single transformation (localization and pulldown assay) or a 1:1:1 ratio for co-transformation (co-localization or BiFC). Leaf disks were excised 2–4 dpi and fluorescence was analyzed in epidermal cells using confocal microscopy.

### Pulldown assay from *N. benthamiana* leaves and peptide identification

*N. benthamiana* leaves were transiently transformed as described above with *A. tumefaciens* (strain GV3101) carrying the vectors pCGFP-RR::SP7ΔSP, pCGFP-RR::SP7 + SP and pCGFP-RR::eGFP[118]. For this purpose, always 2 leaves of 3 individual tobacco plants were infiltrated per construct (6 biological replicates/construct) and after two days whole leave samples were monitored for GFP fluorescence using a binocular to ensure successful transformation. Following this, all biological replicates per construct were combined to generate single pooled samples and total protein extraction and pulldown assays were carried out according to[119] with the following modifications: For immunoprecipitation of GFP-tagged proteins 50 µl of GFP-Trap magnetic beads (Chromotek, Planegg-Martinsried, Germany) were washed 3 times with extraction buffer and incubated for 2 h at 4 °C with 500 µl of a total protein extract containing a protease inhibitor cocktail (1x cOmplete™, Mini, EDTA-free, Sigma-Aldrich, 04693159001). Proteins were eluted from magnetic beads by boiling at 95 °C for 10 min in 50 µl 2x Laemmli sample buffer. For protein identification 25 µl of the eluted protein fraction were loaded on a SDS gel. Protein samples were sliced out as a compact band and were analyzed by mass spectrometry (Toplab GmbH, Martinsried, Germany). Protein samples were reduced with DTT and alkylated with IAA prior to a tryptic digestion (MS grade, Serva) over-night at 37 °C. For nano-ESILC-MS/MS analysis an aliquot of the peptide solution was used. HPLC separation was done using an EASY-nLC1000 (Thermo Scientific) system with the following columns and chromatographic settings: The peptides were applied to a C18 column (Acclaim® PepMap 100 pre-column, C18, 3 µm, 2 cm×75 µm Nanoviper, Thermo Scientific) and subsequently separated using an analytical column (EASY-Spray column, 50 cm×75 µm ID, PepMap C18 2 µm particles, 100 Å pore size, Thermo Scientific) by applying a linear gradient (A: 0.1% formic acid in water, B: 0.1% formic acid in 84% ACN) at a flow rate of 200 nl/min. The gradient used was: 1–30% B in 80 min, 30–60% B in 20 min, 100% B 10 min. Mass-spectrometric analysis was done on a LTQ Orbitrap XL mass-spectrometer (Thermo Scientific), which was coupled to the HPLC-system. The mass-spectrometer was operated in the so-called "data-dependent" mode where after each global scan, the five most intense peptide signals are chosen automatically for MS/MS-analysis. The detected peptide masses were compared with all sequences from *Nicotiana benthamiana* draft genome sequence v1.0.1 at the Solanaceae Genomics Network database (https://solgenomics.net/). The identification threshold was set to p < 0.05. Carbamidomethylation of cysteines was set as fixed modification and oxidized methionine as variable. Only peptides with an Ion score >30 were considered as significant. Proteins identified in the free eGFP samples were subtracted as background. Gene descriptions for identified *Arabidopsis* orthologs were retrieved from TAIR (https://www.arabidopsis.org/tools/bulk/genes/index.jsp).

### Growth phenotyping and analysis of transgenic *A. thaliana* plants

For growth and flowering phenotyping, seeds of respective lines were synchronized over night at 4 °C in the dark and germinated on ½ MS medium for 6 days at 23° ± 1 °C under long day (LD) condition (16 h light/8 h dark). Germinated plants were transferred to soil and analyzed for growth and flower phenotypes after a total of 32 days under LD conditions. For measuring stem elongation of individual plants, plant pictures were taken, and stems were measured using Fiji[120] against a reference scale. For gene expression analysis of *AtFLC*, leaf material of 44-day-old plants was used. For subcellular localization of RiSP2 in *A. thaliana* roots, transgenic RiSP2-GFP plants from T2 or T3 generation were grown on ½ MS medium for 14 days at 23° ± 1 °C under long day (LD) condition (16 h light/8 h dark). Subsequently, GFP fluorescence was monitored using a confocal microscope.

### Quantification of mycorrhizal colonization

WGA-FITZ was employed for immunostaining of fungal structures as described in[118]. Quantification of the mycorrhizal colonization of the immunostained roots was done according to[121]. The calculated parameters were: F%, frequency of mycorrhizal colonization in the root system; M%, intensity of colonization; A%, abundance of arbuscules; and I%, abundance of intraradical hyphae. The observation of the roots for the quantification was done at the confocal microscope.

### Confocal microscopy and image processing

Confocal microscopy images were taken using a Leica TCS SP5 (DM5000) and the LASAF v2.6 software. A HCX PL APO CS 20.0× (NA 0.70) DRY UV (Leica, Wetzlar, Germany) was used at 21 °C. In localization and co-localization studies the fluorescence proteins eGFP (488 nm) and mRFP (561 nm) were excited with an argon and a DPSS 561 laser, respectively. Emission of eGFP was detected from 493–530/550 nm and mRFP from 566 – 670/726 nm. To better discriminate between chloroplast autofluorescence and nuclear bodies in the DsRed and mRFP channels, emissions in far red (675–765 nm) were additionally captured in a separate channel and pseudo colored magenta in merged pictures when necessary. For BiFC analyzes YFP emission was detected from 525–590 nm after excitation at 514 nm. Pictures were processed using Fiji, https://imagej.net/software/fiji/downloads[120].

### Statistical analysis

Gene expression data are shown as boxplots in which the three horizontal lines represent the 3 quartiles, the "x" represents the average and the whiskers reach out to the minimum and maximum values, except for values deemed as outliers, which are shown empty dots, but are included in the statistical analysis. The different parameters of mycorrhizal colonization (F%, M%, A% and I%) are shown as percentages in a grouped barplot in which the error bars represent the standard deviation. On all plots, except for those on Fig. 7 and Supplementary Figs. 19 and 20, the exact values of the individual samples are displayed as a dot on top of the boxes or bars. On Fig. 6e the statistical significance was calculated by performing a one-way ANOVA with a Tukey HSD post-hoc test. The differences among the groups were considered as statistically significant for any $p < 0.05$ and are indicated on the plots with different letters. All other comparisons are of two groups and for those the Shapiro–Wilk test was carried out to check the normal distribution of the data and the Levene's test was used to check the homoscedasticity between each two groups compared. When two groups compared were normally distributed, a two-tailed T-test was carried out in Microsoft Excel applying the corresponding correction depending on whether the homoscedasticity could be assumed or not. When one or both groups were not normally distributed, they were compared with a two-sided Mann-Whitney U test. Significance levels are shown as follows: "ns" (non-significant, $p > 0.05$), "*" (significant $p < 0.05$), "**" (significant $p < 0.01$) or "***" (significant $p < 0.001$). Except for the T-tests, all statistical tests were carried out on the website Statistics Kingdom (www.statskingdom.com) and the box plots were generated with the RStudio software[122]

using the package ggplot2. Sample sizes (n) are specified in the corresponding Figure legends and each n represents material from individual plants.

## Reporting summary

Further information on research design is available in the Nature Portfolio Reporting Summary linked to this article.

## Data availability

The sequences of the SP7-like effectors are available in GenBank with the following accessions: MF521604 (*RiSP7*), MF521605 (*RiSP2*), MF521606 (*RiSP5*), MF521607 (*RiSP31*), MF521608 (*GmarSP1*), MF521609 (*GrosSP1*). *M. truncatula* gene names are from the genome release (Mt4.0v2, http://www.medicagogenome.org/about/project-overview). The RNAseq data generated in this project has been deposited at NCBI with the following accession number BioProject PRJNA1027625. Source data are provided in this paper.

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

## Acknowledgements
We thank Dr. J. Palenzuela (EEZ, CSIC, Spain), Dr. C. Roux (CNRS, Tou-
louse, France) and Dr. S. Roy (Agronutrition, France) for the generous gift
of inocula from *G. margarita* and *G. rosea*. We thank the funding support
of the Future Fields KIT Project "Plants fit for Future" to N.R. We thank F.
Reinicke and J. Wurzinger for their help with the potato and *M. truncatula*
experiments.

## Author contributions
R.B., S.H., D.F-G., and N.R. designed the experiments. R.B., S.H., D.F.-G.,
and M.H. performed the assays. R.B., S.H., D.F.-G., T.L., and N.R. inter-
preted the data. R.B. and N.R. wrote the manuscript with inputs from all
the authors.

## Funding

## Competing interests
The authors declare no competing interests.

## Additional information
**Supplementary information** The online version contains
supplementary material available at

Natalia Requena.

**Peer review information** *Nature Communications* thanks Nicolas
Corradi, Jordana Oliveira and the other, anonymous, reviewers for
their contribution to the peer review of this work. A peer review file is
available.

