## [Peer Review File · Nature Communications]

REVIEWER COMMENTS

Reviewer #1 (Remarks to the Author):

In this study, Betz et al. reported the interactions between AM fungi effectors and splicing factors in plants. The authors performed phylogenetic analysis for the SP7-like family effectors and found they are likely conserved within the Glomeromycotina. The author then analyzed the subcellular localization and interaction partners of several SP7-like effectors by over-expressing them in *N. benthamiana* cells. This led the author to splicing factors SR45, U2AF35b, and U1-70K, among which SP7-like effectors could interact with. Finally, the author showed that ectopic expression of SP7 is accompanied by altered splicing and gene expression in potatoes. The author proposes that SP7-like effectors could hijack or modulate the mRNA processing pathway of the host plant. It is worth noting that similar concepts have already been demonstrated by many other studies. Moreover, I couldn't find convincing data demonstrating the exact scenario as well as the biological/physiological consequences of such interactions between effectors and splicing factors. Overall, I feel the paper falls short in concept and mechanistic details. Below, I summarize my concerns in detail.

1. The localization data of SP7-like proteins is over-interpreted and can be misleading.

The author concluded that these effectors were localized to the nuclear condensates, which represent mRNA processing bodies. The author needs to be very careful about the data obtained in *N. benthamiana* given the protein is highly expressed in such a system. Lots of protein can form bodies or condensate when the concentration is pushed upon a certain threshold, but this doesn't mean the condensation would also occur under physiological conditions (when the fungi interact with the host plant under natural conditions). In addition, lots of essential information is missing about the ability of effector proteins to form the nuclear condensate. For example, could these effectors undergo phase separation *in-vitro* and under what concentrations? Such information would help the reader to appreciate the result in *N. benthamiana*. At the moment, the nuclear condensate observed in *N. benthamiana* can simply be an over-expression artifact. Also, the concept of nuclear condensates or bodies is very vague here, I don't see evidence indicating the nature of such condensates. For example, do these condensates co-localize with transcriptional machinery or mRNAs?

2. The author concluded that SP7-like effectors hijack the mRNA processing pathway. There are a number of issues related to this conclusion. The author mainly performed the Y2H and Co-IP in the *N. benthamiana* by using SP7-like effectors as bait. Although a number of splicing factors were pulled out, there is no evidence indicating that effectors modulate the activity of these proteins/complexes. In line 187-188, the author mentioned, "...SP7-like effectors are recruited to the nuclear condensates...", but there is no evidence to suggest any of the RNA-processing related

proteins localized within the same condensates as the effectors under the physiological condition (when they are not ectopically expressed).

Related to the point above, currently, not only for the SP7-like effectors but also for the majority of RNA processing-related proteins mentioned in this study, there is no sufficient evidence indicating they actually function in the nuclear condensate, as all of them also localize to nuclear plasm. Therefore, the condensate could be non-functional or non-essential.

3. Related to the conclusion that SR45 is the central interaction hub of SP7-like effectors. It is interesting that SR45 can interact with all the SP7-like effectors, but this is not sufficient to claim that SR45 is the organizer of the interactions and, therefore, functions as a potential hub. SP7-like seems can form puncta largely by themselves in *N. benthamiana*, indicating SP7-like proteins may function independent of SR45. The over-expression of SR45 in *N. benthamiana* alters the localization pattern of SP7, but as the authors mentioned, this could be simply an over-expression artifact. Such an assay needs better controls. For example, would the co-expression of other SRs also change the localization patterns of SP7-like effectors? Would the loss of SR45 alter the localization pattern of SP7-like effectors? These should be tested.

4. Related to Figure 2, the author needs to present the actual data instead of a cartoon. The relative abundance of different proteins, repeatability, and controls are critical here. Some data validating these interactions would also be needed.

5. There are several issues related to Figure 7 and the section from lines 257-288.

a. Would ectopic expression of SP7 in potatoes have any physiological or developmental effect, and would such an effect be related to the changes in the alternative splicing as described in this work?

b. In line 267, the author mentioned seven out of eight DEGs were repressed, suggesting the AS isoform may be subject to non-sense mediated decay. Is there any specific evidence for that?

c. Among the SP7-regulated DAS, how many are directly targeted by SP7? The author found 48 genes within DAS genes are targets of SR45 in *Arabidopsis*, but this evidence is rather indirect. Ideally, the author should perform RNA immunoprecipitation and sequencing in the potato or at least in the *N. benthamiana* to have a more convincing picture of it.

d. Would ectopic expression of SP7 alter the gene expression level in general? How many of the altered expressions are associated with altered splicing? These should be carefully described.

Reviewer #2 (Remarks to the Author):

The manuscript by Betz and colleagues demonstrates a role for secreted SP7-like effector proteins from the arbuscular mycorrhizal fungus *Rhizophagus irregularis* in the regulation of alternative splicing of specific host genes in potato. The SP7 homologs forms a small Glomeromycotina specific family. Their results suggest that SP7-like effectors are translocated to host nuclei where they are recruited to nuclear condensates whether they interact with various components of the plant host splicing-machinery. Especially the SR protein SR45 was found as a central interaction hub for SP7-like effectors. In recent years several effectors from pathogenic microbes have been found to target the host splicing machinery to interfere with immune responses. This work now shows that also symbiotic fungi evolved effectors to manipulate host mRNA production. The role of the plant genes affected by the SP7-triggered alternative splicing events remains to be elucidated.

The manuscript is clearly written and the data presented support the conclusions drawn by the authors. This work will be highly interesting for the AM research community as well as plant-microbe research in general.

Below are a few suggestions:

Line 138, as no markers for P-bodies are used, maybe refer to as “likely P bodies”

Perhaps the authors can comment a bit more on the possibility of localization artifacts due to overexpression.

Do the authors also have evidence that SP7-like proteins have the ability to bind RNA?

In the discussion it would be good to expand a bit the examples of microbial effectors that target the host splicing machinery. For example: Tang et al., *Plant Biotech* 2022; Gui et al., *Plant Cell* 2022; Lu et al., *Plant Physiol.* 2023.

Reviewer #3 (Remarks to the Author):

The paper by Betz and collaborators provide evidence that SP7 effectors are abundant in Glomeromycotina, and show their expression during AMF symbioses can modify alternate splicing in the hosts.

The work seems to have used appropriate methods, and I think the results are overall solid. However, I do have concerns about whether some of the statements can be conclusively demonstrated. Specifically, while the authors provide key evidence for their claims/hypotheses,

additional work is needed fully support some of the statements. Overall, the authors need to either provide additional, independent support for the claim that SP primarily affect AS, or significantly town down many of their statements.

General comments:

- The overall message (also see tittle) seem to be that “host gene involved in AS are primarily affected by SP1. However, I do not see any statistical test supporting this claim.

For example, when all genes differentially regulated in the presence of SP1 are considered, how many of these are NOT directly involved in AS, and are these significantly less expressed/regulated than AS genes? I don't seem to find this info in the paper.

- The title of this work make it sound as all AM effectors results in similar outcomes for AS. The authors should thus take other genes into account to test this hypotheses, for example the MycFold effector candidates recently found in *Glomeromycotina*.

Given the massive RNA-seq data available for DAOM 197198, perhaps the authors could dwell more into this important question as a mean to broaden the impact of this study...

- There are a number of claims in the paper that could be investigated further. For example, it is said that AS can lead to the production of STOP codons for some genes, but not to changes in their expression.

It seems odd that AS would be affected but not the overall expression of that gene; any evidence that this is common?

- Regarding changes due to AS: I presume that the “modified protein” (due to AS) will change behaviour? Any evidence of a change in cell localizations/functions following AS?

- The Authors focus on a portion of available genomes.

Specifically, most chromosome-level annotations from “supposed” *R. irregularis* strains (Yildirim 2022, *New Phytologist*), including 4 additional homokaryons and 4 heterokaryons (Sperschneider

2023 Nature Microbiology) have not been investigated, despite knowledge that these vary dramatically in content and structure.

Can those also support the claims made here, including the localization of these genes?

-In particular, looking at the location and expression of SP7s in AMF heterokaryons could lead to interesting findings on how each co-existing genome regulate these effectors. These are fascinating questions, and RNAseq data from multiple conditions are available to test this.

-To follow up on this: the authors state that SP7 are absent in early AMF lineages, however, the Genome of the *Geosiphon pyriformis* (Malar 2021, Current Biology) has not been analysed in this study.

Are SP7s present in this supposed non-mycorrhizal Glomeromycotina, or are there present?

If they are, it could further support the hypothesis made by this species can likely do mycorrhizae, and if not it would provide strong evidence that it cannot.

Either way, this analysis can provide essential insights into the biology and evolution of Glomeromycotina as a whole.

Other specific comments:

L151: It is unclear throughout the MS, which AMF species has been used to test most of this work. I assume its DAOM 197198, but this should be clarified regardless. Accordingly, until similar data has been obtained from other AMF, conclusions should be made by highlighting this obvious caveat.

L186: A figure that show the “predicted model” would be helpful for the naïve reader.

L285: Should test significance. How much more likely is a gene to be affected by AS, as opposed to differential expression? Of all genes affected by SP7, how many are NOT involved ion AS?

Minor comments:

L35 For microbes, they allow them (who allow them?) to remain invisible to the surveillance system of plant... Suggestion: "For example, in microorganisms, the secreted effectors allow them..."

L41 'increasing' showing again, change it or removed it (same word in L37).

L44 missing comma after in planta,

L50 missing dot after 'nucleus'.

L52 what is ca.?

L53 Define NSL signal. This sentence is too long, maybe you can cut from 'pointing to the nucleus...' because is a redundant idea

L54 Try to do a connection with the previous and next sentence, you mentioned the AMF organisms and change to bacteria without a clear transition. Suggestion: "In other symbionts, such as rhizobia bacteria..." or something similar

L61 Define AS (alternative splicing). Several other abbreviations are without its explanation, please present all the nomenclature when first appear.

L75-77 Change it: "A loss of function of a SR protein, the SR45, besides its already known roles in development and abiotic stress, it is also a suppressor of innate immunity in Arabidopsis".

L77-80 This sentence is confusing and have a lot of different information, also it lacks some commas.

L84 RNA recognition motif (RRM) and exon-junction-complex (EJC) are example how you need to address properly other abbreviations.

L81: RNPS1 is a peripheral... it lacks one connector with the previous sentence.

You could add some information about AMF organisms, how they interact with plant (cite the arbuscules and mechanisms of colonization, maybe crucial to the molecules/effectors transport), what they receive and donate... and how that connects with your investigation.

Results

L109 The amplification... and an alignment...

Fig 1d the difference between detected and not detected is missed (I think it was blue and red?)

L128 This sentence could be improved, something like this: We had shown previously that RiSP7 expressed during the symbiosis could be transported to the nucleus of plant' cells as predicted by the presence of an NLS signal. (Is that right? Did you detect the location of that effector or only predicted the NLS?)

Fig 3b The scheme could show the nucleus and cytoplasm regions to be more evident and didactic (such as a sliced cell perspective)

L276 italic to *Arabidopsis*

L280 UTR (untranslated regions) dispenses “regions”, replace to UTR or UTR sequences

Discussions

L328 granula to granule

L335 serve the sequestration -> serve to sequestration

Methods

L446 In bioinformatic methods, please cite which sequences you used as a source for SP7-like prediction.

L489 Go to Gene ontology (GO).

I don't see the IDs for the RNAseq experiment (for example NCBI SRA or Bioproject)

REVIEWER COMMENTS

Reviewer

#1:

In this study, Betz et al. reported the interactions between AM fungi effectors and splicing factors in plants. The authors performed phylogenetic analysis for the SP7-like family effectors and found they are likely conserved within the Glomeomycotina. The author then analyzed the subcellular localization and interaction partners of several SP7-like effectors by over-expressing them in *N. benthamiana* cells. This led the author to splicing factors SR45, U2AF35b, and U1-70K, among which SP7-like effectors could interact with. Finally, the author showed that ectopic expression of SP7 is accompanied by altered splicing and gene expression in potatoes. The author proposes that SP7-like effectors could hijack or modulate the mRNA processing pathway of the host plant. It is worth noting that similar concepts have already been demonstrated by many other studies.

Thank you for taking the effort of revising our manuscript. We agree that there are several other studies that have addressed effectors from pathogenic microbes altering the alternative splicing of their host plants (we cite several of them in our discussion, and now we have included several more). However, this is the first example of a mutualistic symbiotic effector that performs that function and we think this is novel.

Moreover, I couldn't find convincing data demonstrating the exact scenario as well as the biological/physiological consequences of such interactions between effectors and splicing factors.

We agree that we failed to put our results in a physiological context of the role of these effectors altering the AS in plants during mycorrhization. We now provide evidence that these effectors are able to significantly impact on the plant development and in the symbiosis by negatively regulating SR45 in arbuscule containing cells. These are new experiments we have added to the paper which illuminates the model that we propose for SP7-like effectors at the end of the paper (New Figure 8).

Overall, I feel the paper falls short in concept and mechanistic details. Below, I summarize my concerns in detail.

1. The localization data of SP7-like proteins is over-interpreted and can be misleading.

The author concluded that these effectors were localized to the nuclear condensates, which represent mRNA processing bodies. The author needs to be very careful about the data obtained in *N. benthamiana* given the protein is highly expressed in such a system. Lots of protein can form bodies or condensate when the concentration is pushed upon a certain threshold, but this doesn't mean the condensation would also occur under physiological conditions (when the fungi interact with the host plant under natural conditions).

Yes, we agree that several proteins, especially those containing intrinsically disordered domains, like SP7-like effectors, can undergo phase separation in a concentration dependent manner as we discuss in the manuscript. However, this was not the case for our control protein GFP which is also overexpressed nor for shorter versions of SP7 nor for the SR protein SR33 now included in the Supplementary Figure 11.

In addition, results in *A. thaliana* have shown that also SR45 (Fanara et al., JXB 2024) as well as other SR proteins (Fang et al., 2004) localize to nuclear condensates when expressed under the control of their endogenous promoters. Thus, in order to address your comment more specifically, we have now expressed one of these effectors in *A. thaliana* and showed that also there, and not only in *Nicotiana* leaves, SP2 localizes to nuclear condensates in roots and leaves, even in the *hrp* deletion background (see new Supplementary Figure 7 and the picture below further down in one of your other comments, point 3).

We think these results provide further evidence of the nuclear condensates localization of these proteins.

We agree that we cannot know if this is the situation for our effectors under normal physiological conditions since AM fungi cannot be transformed and therefore, we have to rely on heterologous expression *in planta* to investigate the localization of fungal effectors. Nevertheless, what we can clearly observe is that the co-expression of all SP7-like proteins with SR45 in all cases re-localizes all effector proteins to the same nuclear bodies occupied by SR45. This is not the case to other effectors from different families, even if localized in nuclear bodies, as we have previously shown for the CRN1 effector from *R. irregularis*, that despite being located to nuclear bodies, they do not colocalize with SR45 when both proteins are overexpressed (Voß et al., 2018, see picture below). We have now explicitly pointed this out in the results section and included a new picture of this experiment in the Supplementary Figure 11.

Figure 5 from Voß et al., 2018

In addition, lots of essential information is missing about the ability of effector proteins to form the nuclear condensate. For example, could these effectors undergo phase separation in-vitro and under what concentrations? Such information would help the reader to appreciate the result in *N. benthamiana*. Yes we agree, thank you for this suggestion, we are starting experiments in this direction to determine the nuclear condensation ability or not of the effectors, but we think this is not the focus of this manuscript which is the ability to modify the splicing of the plant host cells.

At the moment, the nuclear condensate observed in *N. benthamiana* can simply be an over-expression artifact.

Against this possibility speaks that SP7 when overexpressed alone is not forming nuclear bodies. Only in interaction with SR45 or other interacting proteins known to associate with mRNA does SP7 forms such condensates. However, to obtain more compelling evidence that nuclear condensates localization is the correct localization of these effectors, we have now included an experiment where stable transformed *Arabidopsis* plants are expressing RiSP2-GFP, and the localization of the effector is still in nuclear condensates (New Supplementary figure 7). Although this result still could be due to an overexpression artifact, it is not observed for other overexpressed proteins in *Arabidopsis*. In this direction, the protein HRLP which also localizes in nuclear bodies, only does it when all its intrinsically disordered domains are present, showing that this is not an overexpression artifact (Zhang et al., 2022). This is in line with our results in Figure 4f, where we show that nuclear condensate formation during co-expression of SP7 and SR45 is dependent on the number of repeats of SP7, thus demonstrating that it is not the overexpression what causes the aggregation but the structure of the protein effector, which determines the intensity of the interaction with SR45 (Figure 4e). We have now included a sentence with this information in the Results section.

Also, the concept of nuclear condensates or bodies is very vague here, I don't see evidence indicating the nature of such condensates. For example, do these condensates co-localize with transcriptional machinery or mRNAs? We agree that it is difficult to precisely define the composition of nuclear bodies. Per definition, they are regarded as structures that form and disassemble in a dynamic manner depending on the nucleating proteins present. Our data and the message they convey is that nuclear bodies where SP7-like proteins localize are dynamic structures in which they interact with different components of the mRNA processing machinery from the nascent mRNA till its transport in the cytoplasm. Evidence that some of those bodies are associated to the transcriptional machinery comes from the immunoprecipitation results that show that several components of the transcriptional machinery and splicing are immunoprecipitated by SP7, such as the largest subunit of the RNA polymerase II and RNA export factors, the splicing factors, the polyadenylation factors and many others, which have also been located at nuclear bodies (Saitoh et al., 2014). To provide more evidence of the nature of these bodies we now show also that SP7-like effectors interact with components of the mRNA export machinery (TREX complex, Aly1 protein) also in nuclear speckles (New Supplementary Figure 10), further supporting our model (Figure 8).

Our future experiments will deal with immunoprecipitation experiments and localization of specific mRNAs putatively associated to SP7-like effectors but this is not the scope of this manuscript, which is to show that they interact with well-known splicing factors and are indeed able to induce alternative splicing in plants.

2. The author concluded that SP7-like effectors hijack the mRNA processing pathway. There are a number of issues related to this conclusion. The author mainly performed the Y2H and Co-IP in the *N. benthamiana* by using SP7-like effectors as bait. Although a number of splicing factors were pulled out, there is no evidence indicating that effectors modulate the activity of these proteins/complexes.

We do not agree with the referee here. Although indirect, we show that the constitutive expression of SP7 in plant alters alternative splicing in line with the identified interaction partners.

In addition, we include here a series of new experiments, in which we express SP7 and SP2 in *Arabidopsis*, and show that they recapitulate the phenotype of the Δ SR45 plants with a delayed flowering and a stunted

growth (New Figure 6), suggesting that these effectors act as negative regulators of SR45. Furthermore, we demonstrate that all Arabidopsis orthologues of the alternatively spliced genes identified in potato in response to SP7 that we tested and are known targets of SR45, are also alternatively spliced when SP7 is ectopically expressed in Arabidopsis (Supplementary Figure 17). Altogether these results strongly suggest that the function of SP7-like effectors is to modify the fate of specific mRNA by inhibiting SR45 activity.

In line 187-188, the author mentioned, "...SP7-like effectors are recruited to the nuclear condensates...", but there is no evidence to suggest any of the RNA-processing related proteins localized within the same condensates as the effectors under the physiological condition (when they are not ectopically expressed).

This is right, and it is one of the main constraints when working with this symbiosis, because unfortunately AM fungi are not amenable for stable transformation and thus the only way to perform this experiment is as mentioned above using a proxy solution like expressing the fungal proteins directly *in planta* and interpreting what might be the real function during the symbiosis.

Related to the point above, currently, not only for the SP7-like effectors but also for the majority of RNA processing-related proteins mentioned in this study, there is no sufficient evidence indicating they actually function in the nuclear condensate, as all of them also localize to nuclear plasm. Therefore, the condensate could be non-functional or non-essential.

We agree that the subject of phase separation and nuclear condensates is an emerging field and several aspects are not fully clear yet. However, as shown in the recent paper of Zhang et al., 2022, the nuclear condensate located and RNA binding protein HRLP is only able to complement the mutant phenotype of the *hrlp* mutant when the protein contains the intrinsically disordered domains necessary for localization in nuclear condensates. If they are absent, the protein still localizes to the nucleus but it is non-functional. We think this is a strong argument in favour of the requirement of nuclear condensate formation for the function of these proteins.

We only point out, as it is assumed currently, that nuclear condensates are transient structures that allow proteins to interact in a dynamic manner with certain types of proteins (Banani et al., 2017; Hondele 2019). Moreover, that our effectors localize to those condensates in a concentration dependent-

manner and interact with many proteins related to mRNA processing, suggesting that these effectors could have a role in that mRNA processing. This hypothesis is corroborated by our alternative splicing results.

3. Related to the conclusion that SR45 is the central interaction hub of SP7-like effectors. It is interesting that SR45 can interact with all the SP7-like effectors, but this is not sufficient to claim that SR45 is the organizer of the interactions and, therefore, functions as a potential hub.

Thank you, we think that this is a misunderstanding. We used the word hub to say that SR45 is a central interactor for all SP7-like proteins and that it has a stronger nucleating activity as the other splicing factors (U1-70K and U2AF35b) as it changes the localization of the effectors when co-expressed. But we did not intend to claim that SR45 is the organizer of the interaction with all other proteins that might be located at those nuclear condensates. We have rephrased this title and changed the “hub” word to “core” to avoid misinterpretations.

SP7-like seems can form puncta largely by themselves in *N. benthamiana*, indicating SP7-like proteins may function independent of SR45.

Cells where SP7-like effectors are expressed contain native amounts of NbSR45, thus, we could speculate that those puncta might be also places of SR45 localization, because NbSR45 alone also localizes in nuclear condensates as shown in our new Supplementary Figure 11.

The over-expression of SR45 in *N. benthamiana* alters the localization pattern of SP7, but as the authors mentioned, this could be simply an over-expression artifact.

As we mentioned above and it is known from the literature these proteins change their localization and aggregations in a concentration-dependent manner, and therefore yes, concentration is playing a role in inducing nucleation but not necessarily meaning an artifact. In fact, SR45 and HRLP expressed under its own promoter also localize to nuclear condensates (Zhang et al., 2022; Fanara et al., 2024 JXB). Furthermore, we demonstrate that SP7-like proteins and SR45 do not only co-localize but interact. Also, GFP which is also overexpressed does not change its localization if co-expressed with SR45, and as mentioned above other effectors that localize in nuclear speckles do not re-localize to the SR45 location. For all this reason we do not agree with the referee that this could be simply an overexpression artifact. Nevertheless, we have included a sentence in the results section to comment about the possibility that during symbiosis the size of these condensates might be different due to the amount of protein and consider the possibility that overexpression might have some influence.

Such an assay needs better controls. For example, would the co-expression of other SRs also change the localization patterns of SP7-like effectors?

This question is tricky, because if the SR proteins are interacting with the effectors, they still could do the same as SR45 and thus it would not be a proper control. In contrast, a no-interacting SR protein should not necessarily change the localization of the effectors. But still it could do it through SR45, given that most of them interact with SR45. We tested whether MtSR33 that interacts directly with SP2 and SP5 would be recruiting the proteins to its localization. As shown below, and now also in a new Supplementary Figure 11, this protein is not able to change the localization of SP5 or SP2, thus arguing for a specific effect of SR45.

Would the loss of SR45 alter the localization pattern of SP7-like effectors? These should be tested.

This is an important question but for the moment not possible to answer. We would need to make *Nicotiana* plants lacking SR45. We tried to infiltrate *A. thaliana* *sr45-1* mutants but given that the leaves of these plants were so sick, we could not detect any signs of transformation. This can be due to the fact that SR45 is a suppressor of innate immunity (Zhang et al., 2017) making the plants refractive to transformation by *A. tumefaciens*. As we show in the figure below, this was possible in the WT. We were also able to obtain expression of SP2 in the *hrlp* mutant, and even to show in that genetic background that SR45 co-localizes with SP2 (see below). We have not included this in the manuscript as we think it is distracting, but we could do it if required.

4. Related to Figure 2, the author needs to present the actual data instead of a cartoon. The relative abundance of different proteins, repeatability, and controls are critical here. Some data validating these interactions would also be needed.

The actual data of the Co-IP is given in the Supplementary Table 3, as well as how often a protein was identified in the different Y2H screenings. In addition, the original MS file is available with Nature Portfolio as a pdf file, as well as the RNAseq data. But we think showing the cartoon with the main interactions helps to understand the path of SP7-like effectors. We have moved now the cartoon to the Supplementary Figure 9. In addition, we have added a new Supplementary Figure 10 where we tested and validated the interaction of several of the identified proteins with all SP7 proteins in a yeast two hybrid. In particular, we have analyzed 9 SR proteins (New Supplementary Figure 10). Also, we validated the interaction of the effectors with MtSR45A and MtAly1 by BiFCs. Our data show that some of these effectors are more promiscuous than others and that they do not interact with all SR proteins. This further strengthens our hypothesis that SR45 and U170K and U2AF35 are the core interactors of SP7-like effectors. We have included this in the text.

5. There are several issues related to Figure 7 and the section from lines

257-288.

a. Would ectopic expression of SP7 in potatoes have any physiological or developmental effect, and would such an effect be related to the changes in the alternative splicing as described in this work?

The answer is yes. Expression of SP7 in potatoes changes the development of the plant significantly. We have included now a new figure that shows how SP7-expressing plants are stunted (Supplementary Figure 15). We have also now expressed SP7 and SP2 in Arabidopsis and they clearly showed a delay in flowering and also a stunted phenotype. This is related to the phenotype of the *sr45-1* mutant which is also impaired in flowering and indicates that SP7 proteins might be acting as a negative regulator of the function of SR45 (Figure 6). This hypothesis is supported by a new experiment that we have now included in which we show the relevance of SR45 for the symbiosis. By ectopically expressing SR45 in arbuscule-containing cells we show that the mycorrhizal symbiosis is impaired (Figure 6). Taken together this suggest that SP7 effectors might be negative regulators of SR45 altering in consequence the alternative splicing in cells containing arbuscules. We are not certain yet which one, or several, of the observed changes in AS are related to this phenotype. This is a new project that goes beyond the scope of this manuscript.

b. In line 267, the author mentioned seven out of eight DEGs were repressed, suggesting the AS isoform may be subject to non-sense mediated decay. Is there any specific evidence for that?

We do not have any specific evidence and therefore we have toned down the statement. This hypothesis was based on the fact that RNPS1 the orthologue of SR45 is known also to control the NMD.

c. Among the SP7-regulated DAS, how many are directly targeted by SP7? The author found 48 genes within DAS genes are targets of SR45 in Arabidopsis, but this evidence is rather indirect. Ideally, the author should perform RNA immunoprecipitation and sequencing in the potato or at least in the *N. bethemiana* to have a more convincing picture of it.

We agree, and these are the experiments that are currently under consideration in our group but it is not doable in the time-frame for this revision.

d. Would ectopic expression of SP7 alter the gene expression level in general?

Yes, gene expression is altered, and this data was deposited as mentioned in the accession number of the paper. We have now rewritten this paragraph for more clarity. In general 1,200 genes were regulated among the ca. 30,000 genes of potato.

How many of the altered expressions are associated with altered splicing? These should be carefully described.

As written in that paragraph, only 96 genes are DAS, and from those only 8 were DEGs. We have now written it more clearly.

Reviewer #2 (Remarks to the Author):

The manuscript by Betz and colleagues demonstrates a role for secreted SP7-like effector proteins from the arbuscular mycorrhizal fungus *Rhizophagus irregularis* in the regulation of alternative splicing of specific host genes in potato. The SP7 homologs form a small Glomeromycotina specific family. Their results suggest that SP7-like effectors are translocated to host nuclei where they are recruited to nuclear condensates where they interact with various components of the plant host splicing-machinery. Especially the SR protein SR45 was found as a central interaction hub for SP7-like effectors. In recent years several effectors from pathogenic microbes have been found to target the host splicing machinery to interfere with immune responses. This work now shows that also symbiotic fungi evolved effectors to manipulate host mRNA production. The role of the plant genes affected by the SP7-triggered alternative splicing events remains to be elucidated.

The manuscript is clearly written and the data presented support the conclusions drawn by the authors. This work will be highly interesting for the AM research community as well as plant-microbe research in general.

We thank the referee for taking the time to review our manuscript and the positive comments.

Below are a few suggestions:

Line 138, as no markers for P-bodies are used, maybe refer to as “likely P bodies”

Thank you for the comment, however we had included a marker of P bodies in the supplementary figure 5, but we realised through your comment that we should highlight it also in the main text, what we have done now.

Perhaps the authors can comment a bit more on the possibility of localization artifacts due to overexpression.

We are aware that using the Nicotiana system to express the SP7-like effectors artifacts might appear due to overexpression. However, SR45 has been also shown to localize to nuclear condensates when expressed under its native promoter (Fanara et al., JXB 2024). We have also express them in Arabidopsis (New Supplementary Figure 7), showing the same localization. Furthermore, AM fungi cannot be stably transformed and thus, to analyze the localization of these effectors we have to rely on a heterologous system. As we replied to the first referee, proteins with intrinsically disordered domains are often found in condensates in a concentration-dependent manner. Therefore, yes, it is possible that rather than the localization, the size of these condensates might be different during the symbiosis. We have included a paragraph in the results section to comment on all those possibilities.

Do the authors also have evidence that SP7-like proteins have the ability to bind RNA?

No, we do not have evidence yet. We are currently planning experiments to investigate this aspect and if yes the possible binding sites to the mRNA targets. We are also preparing RNA immunoprecipitation assays with SP7.

In the discussion it would be good to expand a bit the examples of microbial effectors that target the host splicing machinery. For example: Tang et al., Plant Biotech 2022; Gui et al., Plant Cell 2022; Lu et al., Plant Physiol. 2023. Thank you for this comment. We agree that we could have included several other works on the subject. It is done now.

Reviewer #3 (Remarks to the Author):

The paper by Betz and collaborators provide evidence that SP7 effectors are abundant in Glomeromycotina, and show their expression during AMF symbioses can modify alternate splicing in the hosts.

The work seems to have used appropriate methods, and I think the results

are overall solid. However, I do have concerns about whether some of the statements can be conclusively demonstrated. Specifically, while the authors provide key evidence for their claims/hypotheses, additional work is needed fully support some of the statements. Overall, the authors need to either provide additional, independent support for the claim that SP primarily affect AS, or significantly town down many of their statements.

Thank you for taking the time to review our manuscript and for your comments which we are addressing here and in the manuscript

General comments:

- The overall message (also see tittle) seem to be that “host gene involved in AS are primarily affected by SP1. However, I do not see any statistical test supporting this claim.

We think our title was not precise enough. We have changed it now to make it more clear. However, we are unsure what the referee exactly means with this comment. Is he/she meaning that the AS machinery of the plant is affected by the SP7 effector as the sentence says? In this case we do not understand what is meant with the question about the statistical test.

Or is it meant that SP7 is altering the AS of the plant? If this is the case, we show that only a small subset of the potato genome (96) is subjected to alternative splicing in response to SP7 expression, and prove with qRT-PCR that from 11 genes tested, 9 are indeed AS as predicted. We have now also showed that not only these genes are AS in potato plants expressing the second effector RiSP5 as we showed previously but also that when expressed in Arabidopsis, the orthologue genes are also alternatively spliced in response to RiSP7.

For example, when all genes differentially regulated in the presence of SP1 (I think the referee means SP7) are considered, how many of these are NOT directly involved in AS, and are these significantly less expressed/regulated than AS genes? I don't seem to find this info in the paper.

We agree that this paragraph might have been not sufficiently explanatory. The interesting finding is that the modulation of the splicing is not general, but it only affects a specific subset of genes, only few of them being differentially expressed in addition. We had indicated in the manuscript (original line 264 to 266) that from those 96 genes differentially alternatively

spliced, only 8 were differentially expressed, and from those 7 downregulated. Thus, answering your question, from 1227 differentially expressed genes, 1219 are not differentially spliced (see BioProject PRJNA1027625). We have now rewritten this paragraph and added more information to clarify this issue.

- The title of this work make it sound as all AM effectors results in similar outcomes for AS. The authors should thus take other genes into account to test this hypotheses, for example the MycFold effector candidates recently found in Glomeromycotina.

Thank you for this comment. The title was meant to be general to make it more catchy, but seeing the misunderstanding we have now specified it, and we claim that the effectors studied in this paper, the SP7-like family, are involved in modulating alternative splicing. This does not exclude that other effectors might do the same, but we do not claim it.

Given the massive RNA-seq data available for DAOM 197198, perhaps the authors could dwell more into this important question as a mean to broaden the impact of this study...

Thank you for this suggestion but we think to answer that we would need not the kind of RNAseq data that currently exist, but from microdissected arbuscule-containing cells. RNAseq data available have been generated using whole mycorrhizal roots and compared to non-mycorrhizal. This implies that AS changes might go undetected under those conditions if they happen only in a few cells (i.e. arbuscule cells). But even if AS changes would be visible, we could not pin-point them to the specific effect of SP7 expression which is what we are showing and claiming here. Transcriptomic changes were provided in our work and, as mentioned above, the results showed that AS affected only a very small subset of the potato genome.

- There are a number of claims in the paper that could be investigated further. For example, it is said that AS can lead to the production of STOP codons for some genes, but not to changes in their expression. It seems odd that AS would be affected but not the overall expression of that gene; any evidence that this is common?

This is indeed more common as one might think, because not all transcripts having a premature stop codon are sent for non-sense mediated decay (NMD). They might be produced to generate for example transcripts that are either translated at a different speed, or producing shorter protein isoforms

or subjected to different post-translational modifications including regulation by miRNAs (Reddy et al., 2013; Steiger and Brown, 2013; Laloum et al., 2017). This makes that the overall gene expression might not be changed (only the proportion between the isoforms change).

- Regarding changes due to AS: I presume that the “modified protein” (due to AS) will change behaviour? Any evidence of a change in cell localizations/functions following AS?

We show now evidence that expression of SP7 in planta dramatically changes the phenotype of the plants. We have also added an expression experiment in Arabidopsis and show that SP7 plants recapitulate the phenotype of sr45-1 mutant plants, and that several of the transcripts analyzed are also alternatively spliced in this plant. We also have overexpressed SR45 in arbuscule containing cells in Medicago and show that this leads to impaired symbiosis. Therefore, taken all evidence together we speculate that the AS changes induced by SP7 in interaction with SR45 are sufficient to alter the plant cell program of the root cells during symbiosis. However, we do not have yet evidence of the involvement of a specific transcript in this phenotype but are investigating several possibilities such as differential translation efficiency. However, this goes beyond the scope of this paper.

- The Authors focus on a portion of available genomes.

Specifically, most chromosome-level annotations from “supposed” *R. irregularis* strains (Yildirim 2022, New Phytologist), including 4 additional homokaryons and 4 heterokaryons (Sperschneider 2023 Nature Microbiology) have not been investigated, despite knowledge that these vary dramatically in content and structure.

We did only used well annotated genomes. The previous versions were not well assembled and did not properly predict SP7-like effector proteins. We had identified them all by RACE and knew that the versions available (Tisserant et al., 2013; Lin et al., 2014) were containing chimeric versions of SP7-like effectors. We then used the data of Yildirim et al., 2021, as well as the revised version published by Manley et al., 2023 in which they further corrected some annotations and went from 33 to 32 chromosomes. However, the paper of Sperschneider you are mentioning appeared after we have submitted our paper on the 19th of September, thus we could not include it. We have now checked and included this information in the manuscript and pointed out the position, conservation and copy number of SP7-like effectors in those genomes (Supp. Figure 2 and 3, and Source

Data).

Can those also support the claims made here, including the localization of these genes?

Do you mean the chromosome position? If yes, indeed there is a high conservation in the relative position among all strains analyzed, even in the heterokaryons.

-In particular, looking at the location and expression of SP7s in AMF heterokaryons could lead to interesting findings on how each co-existing genome regulate these effectors. These are fascinating questions, and RNAseq data from multiple conditions are available to test this.

Indeed, this could be the subject of interesting experiments. However, this is a functional analysis paper in which we try to demonstrate the mechanism of function of these effectors in planta. However, we will take these comments in consideration for future work.

-To follow up on this: the authors state that SP7 are absent in early AMF lineages, however, the Genome of the *Geosiphon pyriformis* (Malar 2021, Current Biology) has not been analysed in this study. Are SP7s present in this supposed non-mycorrhizal Glomeromycotina, or are there present?

This referee might have missed this information which was included in Supplementary Table 2 (Bioproject PRJNR610605). And no, SP7-like effectors are not present in *Geosiphon pyriformis*.

If they are, it could further support the hypothesis made by this species can likely do mycorrhizae, and if not it would provide strong evidence that it cannot. Either way, this analysis can provide essential insights into the biology and evolution of Glomeromycotina as a whole.

We think this is a statement that is not justified because there is not sufficient data to believe that yet. In particular, because SP7 like effectors are not only absent in *G. pyriformis* but also in other ancient Glomeromycotina that do yes form mycorrhiza. But surely sequencing of new AM species might help to pin point the phylogenetic origin of this effector family.

Other specific comments:

L151: It is unclear throughout the MS, which AMF species has been used to test most of this work. I assume its DAOM 197198, but this should be clarified regardless. Accordingly, until similar data has been obtained from other AMF, conclusions should be made by highlighting this obvious caveat.

This referee missed that this information which was included several times in the manuscript. Yes, it is DAOM197198, but in addition we used two other species (*G. rosea* DAOM194757) and *G. margarita* (BEG34), that were also analyzed throughout the manuscript.

L186: A figure that show the “predicted model” would be helpful for the naïve reader.

Thank you for this suggestion. We have added now a new Figure 8 with a putative model to help understanding the role of SP7-like proteins in AS.

L285: Should test significance. How much more likely is a gene to be affected by AS, as opposed to differential expression? Of all genes affected by SP7, how many are NOT involved ion AS?

We have answered this question in a paragraph above and included in the manuscript.

Minor comments:

L35 For microbes, they allow them (who allow them?) to remain invisible to the surveillance system of plant... Suggestion: “For example, in microorganisms, the secreted effectors allow them...”

Rephrased

L41 ‘increasing’ showing again, change it or removed it (same word in L37).

We do not agree

L44 missing comma after in planta,

corrected

L50 missing dot after ‘nucleus’.

corrected

L52 what is ca.?

It is the abbreviation for the latin form *circa*

L53 Define NSL signal. This sentence is too long, maybe you can cut from ‘pointing to the nucleus...’ because is a redundant idea

corrected

L54 Try to do a connection with the previous and next sentence, you mentioned the AMF organisms and change to bacteria without a clear

transition. Suggestion: “In other symbionts, such as rhizobia bacteria...” or something similar

corrected

L61 Define AS (alternative splicing). Several other abbreviations are without its explanation, please present all the nomenclature when first appear.

corrected

L75-77 Change it: “A loss of function of a SR protein, the SR45, besides its already known roles in development and abiotic stress, it is also a suppressor of innate immunity in Arabidopsis”.

We do not agree

L77-80 This sentence is confusing and have a lot of different information, also it lacks some commas.

corrected

L84 RNA recognition motif (RRM) and exon-junction-complex (EJC) are example how you need to address properly other abbreviations.

Thank you

L81: RNPS1 is a peripheral... it lacks one connector with the previous sentence.

This is explained two sentences above

You could add some information about AMF organisms, how they interact with plant (cite the arbuscules and mechanisms of colonization, maybe crucial to the molecules/effectors transport), what they receive and donate... and how that connects with your investigation.

done

Results

L109 The amplification... and an alignment...

We do not agree

Fig 1d the difference between detected and not detected is missed (I think it was blue and red?)

We still see it, but it might be a problem of the pdf reader the reviewer is using. Because we know that if the program is not Adobe Acrobat sometimes there are problems to see some items of the pdf files.

L128 This sentence could be improved, something like this: We had shown previously that RiSP7 expressed during the symbiosis could be transported to the nucleus of plant' cells as predicted by the presence of an NLS signal. (Is that right? Did you detect the location of that effector or only predicted the NLS?)

This interpretation is wrong. We showed exactly what the sentence says, that the protein localizes to the nucleus when expressed in planta.

Fig 3b The scheme could show the nucleus and cytoplasm regions to be more evident and didactic (such as a sliced cell perspective)

We think the new model clarifies this point

L276 italic to Arabidopsis

done

L280 UTR (untranslated regions) dispenses "regions", replace to UTR or UTR sequences

done

Discussions

L328 granula to granule

We have corrected to granules

L335 serve the sequestration -> serve to sequestration

We do not agree

Methods

L446 In bioinformatic methods, please cite which sequences you used as a source for SP7-like prediction.

All sequences related to our first described SP7 proteins (Kloppholz et al., 2011) that were identified in the genome papers of Tisserant et al., (2013) and Lin et al., 2014, served as query to search for new SP7-like proteins. This is included in the Supplementary Table 2 description and in Materials and Methods.

L489 Go to Gene ontology (GO).

corrected

I don't see the IDs for the RNAseq experiment (for example NCBI SRA or Bioproject)

Please see BioProject PRJNA1027625,

Token:

(<https://dataview.ncbi.nlm.nih.gov/object/PRJNA1027625?reviewer=qgk4h89q31kvb0skg7id97p9n9>)

REVIEWER COMMENTS

Reviewer #1 (Remarks to the Author):

In this resubmitted manuscript, the author addressed most of my previous concerns. I still have a few follow-up questions for the authors.

1. “However, it has been shown that expression of AtSR45 under its natural promoter also drives its localization to nuclear condensates, thus suggesting that nuclear condensate localization per se is not due to an overexpression artifact”

This sounds reasonable but I have to say this is not necessarily true. Multiple tandem copies often occur for T-DNA insertion, which leads to the overexpression of a protein, even if it is driven by the native promoter. This is very common in Arabidopsis. I don't see evidence showing, for example, that the mRNA is expressed at a similar level from the transgene compared with the native condition. Therefore, please turn down the claim here.

2. When talking about the expression of a protein in vivo, please clearly state what promoter is used immediately in all cases. As the author mentioned, condensation can be concentration-dependent.

3. Related to Figure 6 and the section “SP7-like effectors phenocopies the developmental defects of the AtSR45 mutant”.

Fig.6A: what is the relationship between the SP7 expression level and the phenotype? This data should be displayed.

Fig.6B: it is unclear to me at what stage the expression of FLC is examined, but the difference between Col-0 and sr45 or hrlp1 is drastically bigger than what has been reported. Not sure if this data is meaningful.

To claim that SP7 negatively regulates SR45 or Overexpression of SP7 phenocopy the sr45, the author should compare the transcriptome between SP7 overexpression lines and sr45 mutant instead of just looking at FLC. FLC is regulated by lots of genes from many different pathways.

4. Related to “SP7 ectopic expression induces alternative splicing in potato”.

a. It is interesting that only 96 changed alternative splicing events were detected upon SP7 overexpression, but over 1000 gene expression changes were detected. This data seems to suggest the main function of SP7 is not splicing modulation. The author argued that SP7 is a specific splicing regulator, but this seems to contradict the previous proposal by the author that SR45 recruit SP7, as the former is a general regulator of splicing. The author may compare the difference between SP7 overexpression and sr45 or SR45 overexpression lines in Tomatoes to directly validate their core hypothesis that SP7 interacts with SR45 to regulate splicing and, therefore, to regulate plant immunity.

b. The author compared the DAS gene obtained in Tomato to the SR45 RIP targets identified in Arabidopsis. I am not sure such compare is meaningful. What is the assumption behind such a comparison? This should be clearly stated. In addition, the DEGs of SP7 overexpression in Tomato should also be compared with the Arabidopsis SR45 RIP data. An alternative hypothesis here is that SP7 and SR45 affect gene expression mainly independent of splicing. Please clarify.

5. “This result suggests that SP7-like effectors could assemble with the spliceosome already during transcription. In support of this hypothesis, the largest subunit of the RNA polymerase II was immunoprecipitated with RiSP7 (Supplementary Table 3, Fig. 3b). “

Please clearly state, at least in the method section, what is the negative control and how many repeats were performed.

6. “For example, SC35, a central component that links SR proteins to co-transcriptional splicing (ref 54) was also identified as an interactor of RiSP2. “

I'm uncertain about the relevance of ref54 to co-transcriptional splicing. Ref54 is a really old one, and by that time, people refer to splicing as a post-transcriptional event.

Reviewer #2 (Remarks to the Author):

The updated manuscript by Betz et al. reports a novel role for SP7-like effectors of the AM fungus *Rhizophagus irregularis* in the regulation of alternative splicing in the plant host.

In my view, the manuscript is clearly written and the results largely support the drawn conclusions. Several additions have been made, that strengthen the work. One of these additions is the stable expression of SP7-like effectors in Arabidopsis. Similar developmental defects as a result of SP7-like overexpression and SR45 loss-of-function mutants were observed. It would be relevant to include to what extent alternative splicing in these comparisons are similar in Arabidopsis. Now, alternatively spliced genes overexpression of SP7 in potato upon are compared with alternatively spliced genes in the sr45 mutant of Arabidopsis. Given that they now have the overexpression lines of SP7 in Arabidopsis it makes more sense to compare the effect on alternative splicing more directly in Arabidopsis. In supplementary fig. 17, only four out of 11 genes tested by qPCR seemed to be affected by SP7, but it does not become clear whether the observed alternative splicing is similar in the sr45 mutant. Are the other genes not affected?

Perhaps the authors can clarify the difference between SR45 in Fig.3 and SR45A mentioned in Supplementary figure 10.

It would also be good to include data on the expression profile of the SP7-like members during AM symbiosis. In which host cells, i.e. during which stage of the interaction, are these effectors potentially active? For example, for SP7 Zeng et al. (2018) concluded that SP7 is strongly down-regulated as soon as the fungus enters the root, which makes an action in the arbuscule containing cells very unlikely. Does this also hold for the other members? Similarly, in which cell types are the differentially spliced plant genes active during the symbiosis? This is relevant in relation to the proposition made at lines 549-550.

Can the authors indicate/summarize which of the putative interactors from the Y2H are supported by the co-IP experiment and/or the BiFC. Without confirmation with a different technique, true interaction cannot really be concluded.

Reviewer #3 (Remarks to the Author):

The authors addressed most concerns. I am happy to move forward with publication.

Reviewer #4 (Remarks to the Author):

Response to the Referees

Reviewer #1 (Remarks to the Author):

In this resubmitted manuscript, the author addressed most of my previous concerns. I still have a few follow-up questions for the authors.

1. "However, it has been shown that expression of AtSR45 under its natural promoter also drives its localization to nuclear condensates, thus suggesting that nuclear condensate localization per se is not due to an overexpression artifact"

This sounds reasonable but I have to say this is not necessarily true. Multiple tandem copies often occur for T-DNA insertion, which leads to the overexpression of a protein, even if it is driven by the native promoter. This is very common in Arabidopsis. I don't see evidence showing, for example, that the mRNA is expressed at a similar level from the transgene compared with the native condition. Therefore, please turn down the claim here.

Done. We have now toned down our statement in the main text.

2. When talking about the expression of a protein in vivo, please clearly state what promoter is used immediately in all cases. As the author mentioned, condensation can be concentration-dependent.

Done, we have now indicated in the figure legends which promoters were used in all of these cases.

3. Related to Figure 6 and the section "SP7-like effectors phenocopies the developmental defects of the AtSR45 mutant".

Fig.6A: what is the relationship between the SP7 expression level and the phenotype? This data should be displayed.

Thank you, we have now included an additional figure (Supplementary Figure 18) showing that the SP7 expression levels in the different Arabidopsis lines. As it can be seen, there is not a strict correlation between expression levels and the intensity of the phenotype.

Fig.6B: it is unclear to me at what stage the expression of FLC is examined, but the difference between Col-0 and *sr45* or *hrlp1* is drastically bigger than what has been reported. Not sure if this data is meaningful.

The expression of FLC in the *hrlp* mutant (Zhang et al., 2022) was measured as a percentage value of induction, the one in the *sr45-1* mutant was done with RT-PCR and not by qRT-PCR and therefore they are not fully comparable to our expression data which are showing relative values. In addition, they measured expression at very early time points and used the whole plant for those analyses. In our case, as we had written in our previous manuscript version in the M&M, expression was measured only in rosette leaves in plants grown in soil, in the same conditions for all transgenic, mutant and wild type plants. Therefore, the point we do in the manuscript is that ectopic expression of SP7 in Arabidopsis leads to a late flowering phenotype which is accompanied by an induction of the FLC repressor, similar to the phenotype observed in a SR45 loss of function background. In addition, we have now also measured *FT* and *SOC1* in the same samples to further strengthen our point that by interacting with SR45, SP7 leads to the de-regulation of one of the major pathways controlled by SR45 in Arabidopsis. The result shown in new Figure 6 indicate that as expected they are repressed in SP7 expressing lines

similar to *sr45-1* and *hrlp1*. And thus, we consider very meaningful for the interpretation of all the results of the manuscript to include these results.

To claim that SP7 negatively regulates SR45 or Overexpression of SP7 phenocopy the *sr45*, the author should compare the transcriptome between SP7 overexpression lines and *sr45* mutant instead of just looking at FLC. FLC is regulated by lots of genes from many different pathways.

Thank you for this comment. What we meant is that because FLC, and now we also show that FT and SOC1 are similarly regulated between SP7 OE lines and *sr45* mutants, SP7 might be acting by suppressing the SR45 activity and not the opposite. In addition, we show also that SP7 expression in Arabidopsis increases the splicing efficiency of one of the FLC introns, similar to *sr45* and *hrlp* mutants, which could be leading to impaired R loop formation and thus FLC expression. We have included now this new data in the supplementary Figure 18. We do not exclude that SP7 and SR45 could be acting in addition at the transcriptional level. We have introduced this possibility in the main manuscript.

However, we believe that the overlap of transcriptionally regulated genes will be in general low given that two transcriptome analyses of the *sr45* mutant at two different developmental stages (Xing et al., 2015 seedlings and Zhang et al., 2017, inflorescence) gives only a ca.23% overlap. If compared with our root transcriptome analysis this leads to a ca. 11% overlap. We have included this comparison as suggested in the new supplementary Figure 17 and also indicated that in this overlap genes related to immunity functions are identified.

4. Related to “SP7 ectopic expression induces alternative splicing in potato”.
 a. It is interesting that only 96 changed alternative splicing events were detected upon SP7 overexpression, but over 1000 gene expression changes were detected. This data seems to suggest the main function of SP7 is not splicing modulation.

We disagree with the referee in this point for the following reasons:

a) The core interactors of SP7-like proteins are SR45 and the U1 and U2 splicing factors, which are mainly associated with splicing and in particular SR45 with alternative splicing. Furthermore, the GO term analysis of all other interactors suggests mRNA processing and fate.

b) Results from the transcriptome analyses in Arabidopsis from Xing et al., 2015 and Zhang et al., 2017, in which they compared WT vs sr45 mutant at two different developmental stages, show that the percentage of genes undergoing differential splicing as compared to the number of differentially expressed genes is also low (ca. 30%) and from those only ca. 5% are both DAS and DEGs (a similar percentage to our data, 8%). This suggests, in our opinion, that the differentially expressed genes in response to modulation of the SR45 activity could be the result from the differential splicing of upstream regulators. This is also the interpretation of our expression data, since several of the 48 AS that are also SR45 associated genes are major transcriptional regulators.

Nevertheless, we have now included these comparisons in Figure 7 and a scheme in Supplementary Figure 17 and included this discussion in the results and discussion sections.

The author argued that SP7 is a specific splicing regulator, but this seems to contradict the previous proposal by the author that SR45 recruit SP7, as the former is a general regulator of splicing.

SR45, as well as its animal counterpart RNPS1, are not considered general splicing regulators. The picture that it is currently emerging is that these proteins might be “multifunctional splicing regulators that promote correct and efficient splicing of different vulnerable splicing events via the formation of diverse splicing-promoting complexes” Schlautmann et al., 2022. This would explain why the knock-out of SR45 leads to pleiotropic phenotypes. The specificity of their action is likely mediated by accompanying factors such as other SR proteins, in our case we propose this is mediated by the SP7-like proteins (We have also included a small cartoon to explain that in the Supplementary Figure 17).

The author may compare the difference between SP7 overexpression and sr45 or SR45 overexpression lines in Tomatoes (**Potato**) to directly validate their core hypothesis that SP7 interacts with SR45 to regulate splicing and, therefore, to regulate plant immunity.

Thank you for this comment. As mentioned above, we obtained all putative orthologues in Arabidopsis of the potato genes regulated by SP7 and analyzed their regulation in comparison with the data from Xing et al., 2015 and Zhang et al., 2017. Interestingly, although there is some overlap this is very limited, coinciding with the fact that even in the same plant Arabidopsis, the expression of genes at two different developmental stages (flowering and germlings) analyzed between WT and sr45 mutant gave only 25 overlaps of genes regulated in the same direction from a total of only 271 genes regulated in both. Again, this supports the hypothesis that SR45 is not a general splicing regulator and that its function on specific genes is modulated by other developmental/or environmentally regulated factors. The outcome of the transcriptome could be then due to this differential transcript recruitment. What we propose is that SP7 interacts with SR45 only at some of its targets, promoting AS changes to modulate symbiosis. We have included this in our model scheme in Supp. Fig. 17. In this sense, the most significant GO term of the 72 commonly regulated genes is defense (also included in Supplementary Figure 17 with Venn Diagrams).

b. The author compared the DAS gene obtained in Tomato (**Potato**) to the SR45 RIP targets identified in Arabidopsis. I am not sure such compare is meaningful. What is the assumption behind such a comparison? This should be clearly stated.

Because that SR45 is the main interactor of SP7, it is logical to think that SP7 in a way influences the function of SR45, either positively or negatively. Thus, given that the main function of SR45 is the modulation of the alternative splicing, we hypothesized that SP7 overexpression could affect the splicing

of a subset of the targets of SR45 (genes able to be bound and alternatively spliced by SR45), as indeed it was the case. We have rephrased this to make it clearer in the text.

In addition, the DEGs of SP7 overexpression in Tomato (**Potato**) should also be compared with the Arabidopsis SR45 RIP data. An alternative hypothesis here is that SP7 and SR45 affect gene expression mainly independent of splicing. Please clarify.

Thank you for this comment. We think we might have not explained ourselves clearly. When the data set of SR45 RIP is compared to their DEGs in Arabidopsis (Xing et al., 2015 and Zhang et al., 2017), it is shown that only a small percentage of genes (ca. 3 to 7% respectively) are common. This indicates that those genes could be direct targets of SR45 and their expression regulated by the binding of SR45 to the target itself. However, the rest of DEGs (97 to 93% of the genes) could be regulated as a response to the AS of upstream components regulating those genes. As mentioned above, among the SP7-DAS there are several key transcriptional regulators (now shown in New Figure 7). This is the hypothesis that we propose here, that genes that are alternatively spliced in response to SP7 could act as regulators of the expression of the SP7-DEGs identified. This is also one of the argument that Zhang et al., 2017 entertain in their paper, who propose that the AS changes imposed by SR45 on its direct targets further increases the transcriptome complexity of downstream targets. However, we cannot rule out that, in addition, SP7 and SR45 could also regulate gene expression at a different level given that SR45 has been also found associated in a small proportion (10%) to intron-less genes (Xing et al., 2015). One possibility could be for example that SR45 could impact on chromatin remodelling given that some of the interacting partners identified are part of the ASAP complex, and that this hypothesis has been also proposed by others (Milkulski et al., 2022). We have included this hypothesis now in the text.

In addition, to test what the referee suggested, we have obtained the putative orthologues of the potato SP7-DEGs in Arabidopsis (646 genes) and compared them to the RIP SR45 data sets from Xing and Zhang. From those 243 are also SARs but not DAS, indicating that their expression could be directly regulated by the SP7-SR45 complex, independently of splicing. We have now written this possibility in the manuscript and included in the Supplementary Figure 17.

5. “This result suggests that SP7-like effectors could assemble with the spliceosome already during transcription. In support of this hypothesis, the largest subunit of the RNA polymerase II was immunoprecipitated with RiSP7 (Supplementary Table 3, Fig. 3b). “

Please clearly state, at least in the method section, what is the negative control and how many repeats were performed.

We have now added this information also in the Material & Method section and included more details. Previously it was only shown in the corresponding Supplementary Table. The negative control consisted of samples expressing free GFP.

6. “For example, SC35, a central component that links SR proteins to co-transcriptional splicing (ref 54) was also identified as an interactor of RiSP2. “

I'm uncertain about the relevance of ref54 to co-transcriptional splicing. Ref54 is a really old one, and by that time, people refer to splicing as a post-transcriptional event.

Thank you for the comment. We exchanged the reference with a more recent publication that points out the roles of SC35 in co-transcriptional splicing:

Lin S, Coutinho-Mansfield G, Wang D, Pandit S, Fu XD. The splicing factor SC35 has an active role in transcriptional elongation. *Nat Struct Mol Biol.* 2008 Aug;15(8):819-26. doi: 10.1038/nsmb.1461. Epub 2008 Jul 20. PMID: 18641664; PMCID: PMC2574591.

Reviewer #2 (Remarks to the Author):

The updated manuscript by Betz et al. reports a novel role for SP7-like effectors of the AM fungus *Rhizophagus irregularis* in the regulation of alternative splicing in the plant host. In my view, the manuscript is clearly written and the results largely support the drawn conclusions. Several additions have been made, that strengthen the work. One of these additions is the stable expression of SP7-like effectors in Arabidopsis. Similar developmental defects as a result of SP7-like overexpression and SR45 loss-of-function mutants were observed.

It would be relevant to include to what extent alternative splicing in these comparisons are similar in Arabidopsis. Now, alternatively spliced genes overexpression of SP7 in potato upon are compared with alternatively spliced genes in the sr45 mutant of Arabidopsis.

Thank you for this comment. Actually, we think there is a misunderstanding here, because we did not compare the alternative splicing events of potato in response to OESP7 with those of the sr45 mutant but with the genes associated with SR45 that were identified by immunoprecipitation. This is because we think that depending on the proteins that associate with SR45, the AS targets will be determined. In our case SP7, but these are surely different to those in response to stresses or to developmental cues.

Given that they now have the overexpression lines of SP7 in Arabidopsis it makes more sense to compare the effect on alternative splicing more directly in Arabidopsis.

Thank you for this suggestion. This will be possible with RNAseq from Arabidopsis expressing SP7 which we do not have yet. Therefore, we limited our comparison to the selected genes and included your suggestions. See below.

In supplementary fig. 17, only four out of 11 genes tested by qPCR seemed to be affected by SP7, but it does not become clear whether the observed alternative splicing is similar in the sr45 mutant. Are the other genes not affected?

From the 11 genes that we tested in potato, only 8 with had an orthologue in Arabidopsis with several predicted alternative spliced forms. We now measured all those 8, and for 7 of them we could show AS in response to SP7. Furthermore, we have measured their AS in the sr45 mutant and found that from those 7, 4 are AS in the same direction as in response to SP7, and 1 in the opposite direction. The other two are either not significantly regulated or not at all (New Supplementary Figure 20).

Perhaps the authors can clarify the difference between SR45 in Fig.3 and SR45A mentioned in Supplementary figure 10.

Thank you for this comment. We have now included a phylogenetic tree of SR protein family of different plant species where SR45A is also included (Supplementary Figure 11 and 12) and indicated there which of those members were identified as effector interaction partner of SP7-like proteins.

It would also be good to include data on the expression profile of the SP7-like members during AM symbiosis. In which host cells, i.e. during which stage of the interaction, are these effectors potentially active? For example, for SP7 Zeng et al. (2018) concluded that SP7 is strongly down-regulated as soon as the fungus enters the root, which makes an action in the arbuscule containing cells very unlikely. Does this also hold for the other members?

Thank you for this comment. In fact, there have been several papers wrongly ascribing the expression of SP7 due to the fact that until recently the annotation of the *R. irregularis* genome was very incomplete, and the paralogues were not clearly identified. However, the resequencing and new annotation of the *R. irregularis* by the Corradi group (Yildirim et al., 2022 and Manley et al., 2023), clearly have shown the existence of these paralogues (see our Figure 1 and corresponding supplementaries). Therefore, read mapping in previous papers (Zeng et al., 2018) is uncertain. In addition, data from other groups (Kamel et al., 2017) and ours (Kloppholz et al., 2011) have shown that even if expressed in other tissues, SP7 and other SP7-like effectors are induced during planta colonization. Moreover, the recent paper of single nuclei transcriptomics of mycorrhizal roots clearly shows that SP7 is expressed in the cluster corresponding to arbuscule containing cells (Serrano et al., 2024). The same cluster also contains the MycFold effectors, which were shown by *in situ hybridization* to be located in arbuscules (Teulet et al., 2023). We have done laser capture microdissection and also corroborated that SP7 is induced in arbuscules (ARB) as compared with colonized neighbour cells (NAC), but this is included in another manuscript we are currently writing (see picture below). Taken together, we are convinced that during the *in planta* phase, SP7 is mainly expressed in arbuscules. We have indicated this now more clearly in the manuscript.

Similarly, in which cell types are the differentially spliced plant genes active during the symbiosis? This is relevant in relation to the proposition made at lines 549-550.

As shown above, we think that this makes more likely that the function of SP7 is exerted in arbuscule containing cells and the changes that it induces in splicing and gene expression are mainly restricted to those cells.

Can the authors indicate/summarize which of the putative interactors from the Y2H are supported by the co-IP experiment and/or the BiFC. Without confirmation with a different technique, true interaction cannot really be concluded.

Thank you for the suggestion. All identified putative interaction partners from our screen approaches were shown in Supplementary Table 3. There (sheet "core interactors"), we also indicated which

proteins were retrieved by more than one effector member in the Y2H screens or in both screens (Y2H and SP7 Co-IP screens). But indeed, we had not summarized which of the proteins were additionally confirmed with either direct Y2H or BiFC assays in our manuscript. To make it more comprehensible for the reader, we have now included this information in a new Supplementary Figure 10 that summarizes all putative interaction partners that were identified more than once in different screens and which proteins were tested in direct interaction assays. However, although true interaction for the whole data set could be only be concluded when using different techniques, the sheer amount of identified putative plant interactors with a role in RNA processing (Supplementary Table 3 sheet “Go Term Enrichment” and Supplementary Figure 9) in each independent screen strongly and in particular many SR protein family members suggests a complex multilayered effector-plant RNA processing protein interaction network.

We thus focused on SR45 that was identified in all untargeted screens as well as the SR45 interacting core splicing proteins U2AF and U1-70k for in depth interaction analysis in the manuscript.

Reviewer #3 (Remarks to the Author):

The authors addressed most concerns. I am happy to move forward with publication.

Thank you!

Reviewer #4 (Remarks to the Author):

Thank you!

REVIEWERS' COMMENTS

Reviewer #1 (Remarks to the Author):

The author addressed most of my previous concerns. I am happy to move forward.

Reviewer #2 (Remarks to the Author):

The authors have sufficiently addressed the concerns, recommending publication.